# A variant NuRD complex containing PWWP2A/B excludes MBD2/3 to regulate transcription at active genes

Tianyi Zhang[1], Guifeng Wei [1], Christopher J. Millard [2], Roman Fischer [3], Rebecca Konietzny[3,4], Benedikt M. Kessler [3], John W.R. Schwabe [2] & Neil Brockdorff[1]

Transcriptional regulation by chromatin is a highly dynamic process directed through the recruitment and coordinated action of epigenetic modifiers and readers of these modifications. Using an unbiased proteomic approach to find interactors of H3K36me3, a modification enriched on active chromatin, here we identify PWWP2A and HDAC2 among the top interactors. PWWP2A and its paralog PWWP2B form a stable complex with NuRD subunits MTA1/2/3:HDAC1/2:RBBP4/7, but not with MBD2/3, p66α/β, and CHD3/4. PWWP2A competes with MBD3 for binding to MTA1, thus defining a new variant NuRD complex that is mutually exclusive with the MBD2/3 containing NuRD. In mESCs, PWWP2A/B is most enriched at highly transcribed genes. Loss of PWWP2A/B leads to increases in histone acetylation predominantly at highly expressed genes, accompanied by decreases in Pol II elongation. Collectively, these findings suggest a role for PWWP2A/B in regulating transcription through the fine-tuning of histone acetylation dynamics at actively transcribed genes.

---

[1] Developmental Epigenetics, Department of Biochemistry, University of Oxford, Oxford OX1 3QU, United Kingdom. [2] Leicester Institute for Structural and Chemical Biology and Department of Molecular and Cell Biology, University of Leicester, Leicester LE1 7RH, United Kingdom. [3] Target Discovery Institute, Nuffield Department of Medicine, University of Oxford, Roosevelt Drive, Oxford OX3 7FZ, United Kingdom. [4] Agilent Technologies, Hewlett-Packard-Str. 8, 76337 Waldbronn, Germany. These authors contributed equally: Tianyi Zhang, Guifeng Wei. Correspondence and requests for materials should be addressed to N.B. (email: neil.brockdorff@bioch.ox.ac.uk)

Transcriptional regulation by the epigenome is conferred through the interplay between the writers, erasers, and readers of histone modifications[1]. Readers recognise modifications, such as phosphorylation, acetylation, and methylation through specialised protein domains. H3 Lysine 36 tri-methylation (H3K36me3) is a conserved posttranslational histone modification associated with gene transcription and enriched on the bodies of active genes. Readers of H3K36me3 have been found to be involved in DNA replication and repair, DNA methylation, transcriptional elongation, alternative splicing, and repression of spurious initiation[2]. H3K36me3 binding is conferred by either Chromo- or PWWP PTM-interacting modules that are present in many chromatin-associated proteins.

One of the best characterised readers of H3K36me3 is the Eaf3 subunit of the Rpd3S histone deacetylase complex in *Saccharomyces cerevisiae*. The yeast HDAC complex is recruited to the gene bodies of actively transcribed genes through binding of Eaf3 to H3K36me3[3–5]. Histone acetylation promotes a chromatin environment permissible to transcription initiation and is found at gene promoters[6]. Recruitment of the yeast HDAC complex by H3K36me3 enforces a low level of histone acetylation over the coding regions to repress cryptic transcription initiation[3–5]. Equivalent pathways in mammalian organisms have not been identified.

HDAC proteins HDAC1/2 are the mammalian homologs to the yeast Rpd3, and form many types of Class I complexes, including, NuRD, Sin3, MIDAC, and CoREST[7,8]. Conventionally, HATs and HDACs were proposed to act antagonistically to either promote or repress gene transcription through the establishment of hyper or hypo-acetylated chromatin states. Although, HDACs have traditionally been characterised as transcriptional co-repressive complexes, they have also been implicated in gene activation[9]. Over the past decade, an emerging view suggests that the dynamic turnover of acetylation through the concerted activity of both HATs and HDACs, rather than stable hyperacetylation, is important for gene activation[6]. This is supported by genome-wide ChIP-seq data, which show that both HATs and HDACs bind active genes, positively correlating with gene transcription[10]. Gene modulation by HDACs may be highly context specific, as depletion of HDAC1 or treatment of cells with HDAC inhibitors (HDACi) lead to roughly equal numbers of genes being upregulated and downregulated[9,11]. Notably, studies employing HDACi treatment have linked HDAC inhibition to Pol II elongation blockade, through crosstalk with elongation factors, such as BRD4 and P-TEFb[12–14]. These results indicate a role for HDACs in gene activation through promoting transcriptional elongation. However, as HDACi are broad spectrum inhibitors of both Class I and Class II HDACs, the precise roles of different HDAC complexes in transcriptional control requires further investigation. Moreover, how HDACs are recruited to specific targets to modulate acetylation turnover is not fully elucidated.

Our study focuses on the nucleosome remodelling and deacetylase (NuRD) complex, one of the mammalian Class I HDAC complexes. The NuRD complex was the first chromatin-associated complex found to possess two enzymatic activities, histone deacetylation mediated by the HDAC1/2 subunits, and ATP-dependent nucleosome remodelling by the CHD3/4 subunits[15–17]. In recent years, multiple studies have found that the deacetylase core of NuRD (MTA1/2/3, HDAC1/2, and RBBP4/7) forms a stable subcomplex[18–20], and that components such as MBD2/3, and CHD3/4 are often substoichiometric or transiently interacting[20]. Factors controlling NuRD assembly are not well understood, and how the complex exerts its enzymatic and transcriptional regulatory activities in varying genetic contexts remains to be explored.

In this study, using an unbiased proteomic approach to identify interactors of H3K36me3, we find three HDAC associated proteins enriched on H3K36me3-modified nucleosomes, HDAC2, RBBP7, and PWWP2A. PWWP2A is an evolutionarily conserved protein that contains a PWWP domain, a module known to bind H3K36me3. PWWP2A has been observed in HDAC interactome studies[21], but its biochemical and functional role in HDAC biology remains uncharacterised. Here we find that PWWP2A co-purifies with components of the NuRD complex, associates stoichiometrically with MTA1/2/3, HDAC1/2, and RBBP4/7 (the deacetylase subunits of NuRD) and is mutually exclusive with MBD2/3, p66α/β, and CHD3/4. By ChIP-seq, we detect that PWWP2A and PWWP2B are most enriched at genes with high expression level, sharing many targets with NuRD subunits. Finally, we show that deletion of PWWP2A/B triggers an increase levels of H3K9ac and H3K27ac as well as a decrease in RNA Pol II elongation in mESCs, suggesting that this protein may be involved in the regulation of Pol II dynamics and transcription in vivo.

## Results

**Identification of PWWP2A as an H3K36me3 reader**. Trimethylation of H3K36 is a highly conserved modification associated with active transcription. Unlike many other active histone modifications which are enriched at gene promoters, H3K36me3 is highly correlated with the gene bodies of transcribed genes. To gain more insights into the function of this modification, a proteomic screen was performed to identify H3K36me3-binding nuclear proteins (Fig. 1a).

We used recombinant unmodified and H3Kc36me3-modified dinucleosomes as bait to pull on unlabelled and SILAC labelled nuclear extract to identify nuclear proteins preferentially enriched on H3Kc36me3-modified nucleosomes (Fig. 1a, b and Supplementary Fig. 1a) similar to Bartke et al.[22]. H3Kc36me3 mimics were generated as previously described[23] (Fig. 1b and Supplementary Fig. 1b). Proteins which were enriched on H3Kc36me3 nucleosomes in two biological replicates (forward and reverse pull-downs) are shown in Fig. 1c. Aside from the known H3K36me3-binding protein MSH6, three of the most enriched proteins were PWWP2A, HDAC2, and RBBP7. HDAC2 and RBBP7 form the core of Class I deacetylase complexes, and PWWP2A has previously been observed to interact with HDAC1/2 complexes in human T cells[21]. Importantly, PWWP2A has a known H3K36me3-interacting PWWP domain at the C-terminus of the protein.

PWWP2A is conserved in chordates, and a gene duplication event during early vertebrate evolution gave rise to a paralog PWWP2B (Supplementary Fig. 1c,d). Both the *PWWP2A* and *PWWP2B* genes undergo alternative splicing and polyadenylation to produce multiple protein isoforms, a long isoform and short isoforms, which lack the PWWP domain (Supplementary Fig. 1c). The aromatic cage residues that bind H3K36me3 in the PWWP domains of MSH6 and DNMT3B are also conserved in PWWP2A/B (Supplementary Fig. 1e-g). To determine whether the H3K36me3 binding is dependent on the PWWP domain, mouse embryonic stem cells stably expressing Flag-2xStrepII (FS2) tagged form of full-length PWWP2A and a truncation lacking the PWWP domain (PWWP2AΔPWWP) were generated (Fig. 1d). By ChIP-qPCR, we observe PWWP2A binding to both the promoters and gene bodies of three active genes in mESCs (*Nanog*, *Acat2*, and *Neil2*), with higher binding at the gene bodies which are enriched for H3K36me3 (Fig. 1e). Deletion of the putative H3K36me3-binding PWWP domain, strongly abrogates PWWP2A binding at the gene bodies of *Nanog*, *Acat2*, and *Neil2*. From this, we infer that the PWWP

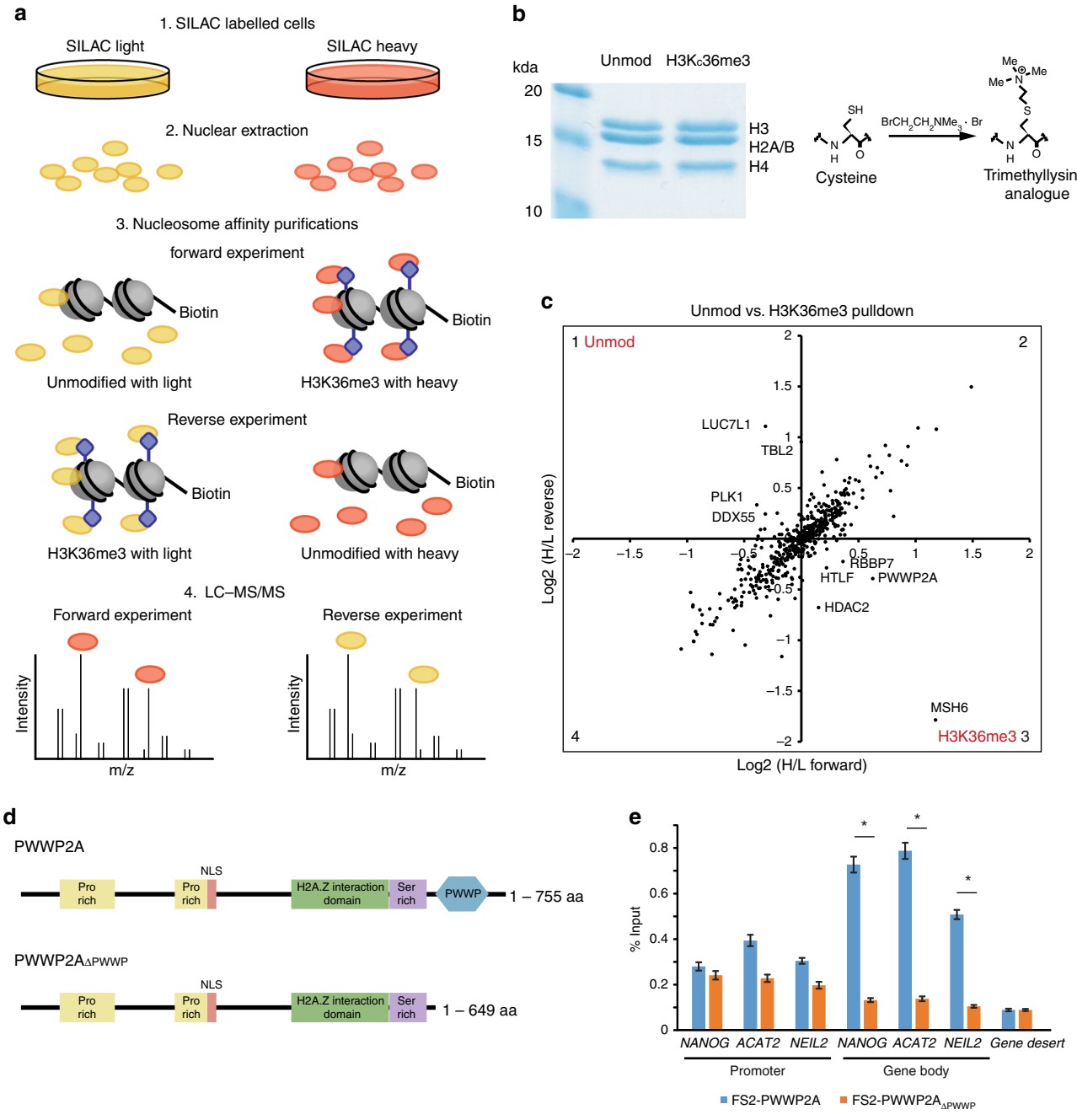

**Fig. 1** Identification of PWWP2A as an H3K36me3 binding protein. **a** SILAC nucleosome affinity purifications were performed as previously described in Bartke et al.[22]. HeLa cells were labelled with heavy or light SILAC media (1) and nuclear proteins were extracted (2). Pairwise nucleosome affinity purifications were performed as follows (3): light and heavy nuclear extracts were mixed with unmodified and H3K36me3-modified recombinant nucleosomes respectively in the Forward experiment and vice-versa in the Reverse experiment. The paired nucleosome affinity purifications in the forward experiment were pooled for LC–MS/MS (4), and the same for the reverse experiment. **b** Unmodified and H3Kc36me3-modified nucleosomes were generated as described in Dyer et al.[51]. H3K36C purified histones were alkylated to completion to generate site-specific cysteine to generate a trimethyllysine analog, as described[23]. **c** Identification of proteins enriched on unmodified vs. H3K36me3-modified nucleosomes. Proteins enriched on H3K36me3-modified nucleosomes are found in quadrant 3. **d** Domain structure of the PWWP2A protein. PWWP2A has a C-terminal PWWP domain (blue), the rest of the protein is highly disordered and contains two proline-rich (yellow) and one serine-rich (purple) region. PWWP2A was recently found to contain a H2A.Z-interacting region[33] (green). **e** ChIP-qPCR of FS2-PWWP2A and FS2-PWWP2AΔPWWP at the promoters and gene body regions of three genes expressed in mESCs, and a gene desert which does not harbour any active or repressive chromatin modifications. The error bar represents s.e. m. from four biological replicates, and significance was calculated by Student's $t$-test, * indicates $p$-value <0.05

domain of PWWP2A is required for its binding to chromatin containing H3K36me3.

**PWWP2A/B defines a variant of the NuRD complex**. To further understand the physiological roles of PWWP2A, we next sought to characterise the PWWP2A interactome and its association with HDAC1/2. In mammalian organisms, HDAC1/2 can be found in HDAC complexes, including NuRD, Sin3, MIDAC, and CoREST[7,8]. Purification of PWWP2A-associated proteins from HeLa and mESCs yielded NuRD subunits MTA1/2/3, HDAC1/2,

and RBBP4/7 under high salt (500 mM NaCl) conditions (Fig. 2a, b and Supplementary Fig. 2a,b). PWWP2A also co-purified four transcription factors ZFP423, ZFP521, TIF1β, SALL4 and a chromatin binding protein BEND3 which have previously been known to associate with the NuRD complex[24–26] (Fig. 2a).

Notably, although a large number of peptides were detected from MTA1/2/3, HDAC1/2, and RBBP4/7, no peptides were detected from the NuRD components MBD2/3, CHD3/4, or p66α/β (Fig. 2a and Supplementary Fig. 2b). We confirmed that PWWP2A co-immunoprecipitates MTA1 and HDAC1/2, but not CHD4 or MBD3, using two independently derived clones stably expressing different levels of FS2-PWWP2A, implying that PWWP2A exclusively interacts with the deacetylase subcomplex of NuRD (Fig. 2b). Furthermore, we performed fractionation of nuclear extract from E14 mESCs using size exclusion chromatography and immunoblotted for CHD4, MBD3, HDAC1, MTA1, and Flag (PWWP2A) (Fig. 2c). All five proteins eluted in very high molecular weight fractions, as previously observed for NuRD complex components[17]. HDAC1 and MTA1 elute over multiple fractions, possibly indicating their presence in complexes of varying molecular weights. PWWP2A elutes in a high molecular weight complex over many fractions containing MTA1 and HDAC1 (fractions 15–25), and with minimal overlap with MBD3 (fractions 7–15) (Fig. 2c). We infer that fractions 7–15, which contain CHD4, MTA1, HDAC1, and MBD3 but not PWWP2A, represent the canonical NuRD complex, which runs at a higher molecular weight than the PWWP2A:MTA1:HDAC1 complex (fractions 7–15). CHD4, which is known to have NuRD-independent functions, elutes in two peaks, one coinciding with the canonical NuRD complex (fractions 7–15), and another presumably NuRD-independent peak (fractions 19–25).

Through expression of truncated forms of PWWP2A (Fig. 2d), we determined that a region in the N-terminal half of PWWP2A, residues 148–373 is necessary and sufficient to mediate the interaction with MTA1 and HDAC1 and occlude MBD3. This region has no obvious domain structure but contains a proline-rich region and is highly conserved between PWWP2A, PWWP2B and their homologs from various species. We confirmed that both the short isoform of PWWP2A and PWWP2B also co-immunoprecipitate MTA1 and HDAC1 but not CHD4 or MBD3 (Supplementary Fig. 2c,d). Additionally, both paralogs co-purify with each other indicating that at least two copies of PWWP2A/B exist in the same protein complex (Supplementary Fig. 2b, d). From our MS analysis, under highly stringent conditions, PWWP2A and PWWP2B did not stably associate with any other complexes aside from the NuRD deacetylase subunits, suggesting that in vivo, PWWP2A/B functions exclusively in the context of the NuRD deacetylase sub-complex (Fig. 2a and Supplementary Fig. 2d).

**PWWP2A forms an enzymatically active complex with MTA: HDAC.** Our previous studies have demonstrated that the co-repressor protein MTA1 serves as a scaffold for the core NuRD complex[18,19]. MTA1 dimerizes through its ELM2 domain and assembles HDAC1/2 and RBBP4/7 into the complex. To investigate the interaction of PWWP2A with the core NuRD complex, PWWP2A1–373 was co-expressed in HEK293F cells, and co-purified with FLAG-MTA1 162-546, HDAC1, and RBBP4 (Fig. 3a, b). PWW2A1–373 forms a stable complex with MTA1, HDAC1, and RBBP4 with an apparent stoichiometry of 2:2:2:2 as determined by SDS-PAGE visualisation.

In contrast, when PWWP2A1–373 was co-expressed with FLAG-MTA1 162-354, HDAC1, and MBD3, two distinct complexes were formed. One containing PWWP2A:MTA1: HDAC1 and a second complex containing MBD3:MTA1:

HDAC1. The two complexes could be clearly separated using size exclusion chromatography (Fig. 3c), supporting the mass spectrometry data which suggests mutually exclusive binding of PWWP2A and MBD3 to the MTA1:HDAC1 complex. Since this is a biochemically pure system, it would suggest that there is direct steric overlap between the binding sites for PWWP2A/B and MBD2/3 that prevents both proteins binding at the same time. Given that PWWP2A does not interact with HDAC1 or RBBP4 in the absence of MTA1 (Supplementary Fig. 3a), we conclude that PWWP2A1–373 and MBD3 are most likely interacting directly with MTA1. This is in line with our mass spectrometry results which find that the most abundant PWWP2A interactor in vivo is the MTA1/2/3 family of proteins (Fig. 2a and Supplementary Fig. 2b,d).

Our results show that residues 1–373 of PWWP2A are sufficient for interaction with the NuRD subcomplex. A sequence alignment of PWWP2A and PWWP2B from various species was used to compare these proteins. It was notable that there is relatively little conservation within this region other than a stretch of 66 residues (264–329) which has 45% identity and 68% similarity (Supplementary Fig. 3b). Importantly, a short motif (R/QPRQV/L) in this region is very similar to a conserved sequence in MBD2/3. This led us to speculate that this motif may form part of the binding interface to HDAC1:MTA1 since it could explain the mutually exclusive binding between PWWP2A and MBD3. Measurements by fluorescence anisotropy show that a PWWP2A peptide containing this motif binds with an affinity of between 20 and 100 micromolar to HDAC1:MTA1 (Supplementary Fig. 3c). It is likely that adjacent regions also contribute to the binding allowing a stable complex to be formed. Finally, analysis of the MTA1:HDAC1:PWWP2A complex reveals that this complex retains histone deacetylase activity to a level comparable to that of MTA1:HDAC1 with or without MBD3 (Fig. 3d).

**PWWP2A/B is enriched at highly expressed genes in mESCs.** Having established that PWWP2A/B forms a complex with the deacetylase core of NuRD, we set out to explore the function of the PWWP2A/B-deacetylase complex in vivo. ChIP-seq was performed to map the genome wide localisation of PWWP2A and PWWP2B in mESCs (Fig. 4a–e). PWWP2A was enriched over the gene bodies of active genes colocalising to a high degree with H3K36me3 (Fig. 4a, b, d, e). ENCODE transcriptome data for this cell line was used to divide genes equally into four groups with high, intermediate, low, or no expression (Fig. 4c). PWWP2A exactly mirrors the pattern of occupancy as H3K36me3 at highly, intermediately, and lowly expressed target genes genome-wide (Fig. 4a, d). Binding of PWWP2A to chromatin was highly dependent on the presence of the PWWP domain as the short isoform and a construct lacking the PWWP domain lose their gene body localisation entirely and retain only a small level of promoter enrichment (Fig. 4e and Supplementary Fig. 4a,b). A representative example of our ChIP-seq tracks at the *Mta2-Tut1* locus clearly shows that PWWP2A co-localises with H3K36me3 over the gene bodies of active genes, and this pattern is lost in constructs lacking the PWWP domain (Fig. 4e). Interestingly, the profile of PWWP2B binding differed from that of PWWP2A, with PWWP2B showing promoter and enhancer enrichment in mESCs (Fig. 4d, e and Supplementary Fig. 4c,d,f). Both PWWP2A and PWWP2B are enriched at active chromatin marked by H3K4me3 and H3K36me3 (Fig. 4d), and positively correlates with gene transcription, with the majority of PWWP2A and PWWP2B targets being the most highly expressed genes (Fig. 4c, d).

In parallel, we generated ChIP-seq data for endogenous HDAC1 and MTA2 in mESCs, and also re-analysed published

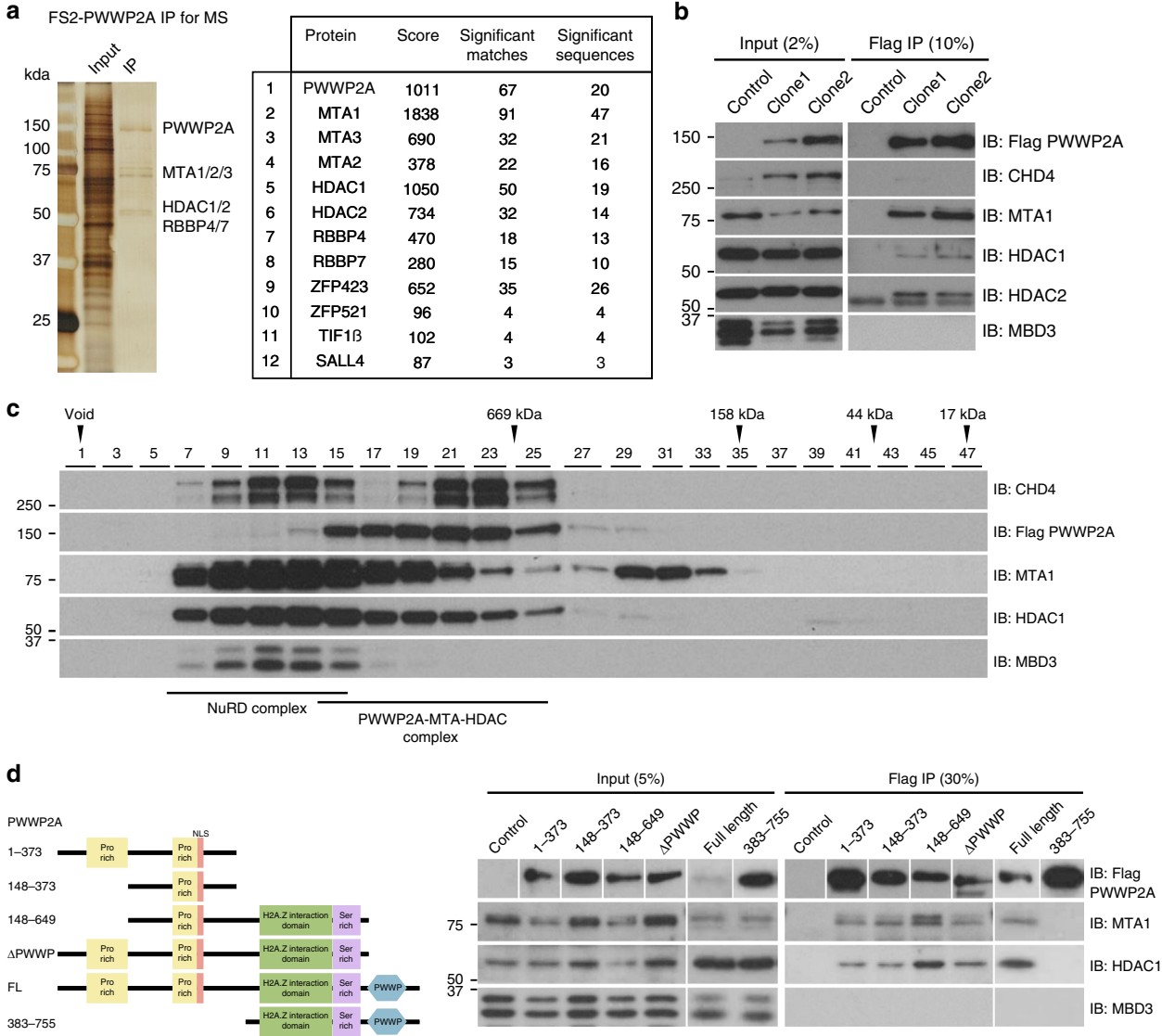

**Fig. 2** PWWP2A interacts with deacetylase subunits of the nucleosome remodelling and deacetylase complex. **a** Silver stain of FS2-PWWP2A-associated proteins from mESCs, and list of most significant interactors detected by LC–MS/MS and identification by Mascot (1% FDR). **b** PWWP2A co-immunoprecipitates NuRD subunits MTA1 and HDAC1/2 but not CHD4 or MBD3 in mESCs. **c** Fractionation of E14 mESC nuclear extract followed by size exclusion chromatography (Superose 6) and immunoblot detection of FS2-PWWP2A (88 kDa) and NuRD subunits CHD4 (isoform1 218 kDa), MTA1 (81 kDa), HDAC1 (55 kDa), and MBD3 (isoform1 32 kDa, isoform2 28 kDa). Fractions 7–15 contain the holo-NuRD complex (CHD4, MTA1, HDAC1, MBD3), while fractions 15–25 contain the PWWP2A:MTA1:HDAC1 complex. NuRD-independent CHD4 elutes in fractions 19–25. **d** Truncated constructs of FS2-PWWP2A were used to assay the region of PWWP2A that binds MTA1 and HDAC1. PWWP2A1–373, 148–373, 148–649, ΔPWWP co-immunoprecipitate MTA1 and HDAC1, while PWWP2A 383–755 does not. PWWP2A148–373 contains the region in PWWP2A that is both necessary and sufficient to mediate the MTA1 and HDAC1 interaction

ChIP-seq data for MBD3 and CHD4. HDAC1, MTA2, MBD3, and CHD4 are also predominantly associated with active genes (Fig. 4c, d). HDAC1, MTA2, MBD3, CHD4, and PWWP2B peaks can be found at active gene promoters, enhancers, and the 5′ proximal gene body regions, while PWWP2A is primarily associated with gene bodies, albeit with some binding at poised enhancer regions (Supplementary Fig. 4f). It is noteworthy that while our biochemical experiments demonstrate that MTA1:HDAC1:PWWP2A form a stable stoichiometric complex, ChIP-seq analysis indicates distinct chromatin binding profiles for MTA2 and HDAC1 (predominantly over gene promoters) and PWWP2A (predominantly over gene bodies). It has been shown previously that both gene promoter and gene body regions are subject to deacetylation by HDAC complexes in vivo[27], but by

ChIP HDAC1 shows much stronger enrichment at promoters than gene bodies. We surmise that the majority of MTA and HDAC1/2 proteins are recruited to gene promoters as part of the canonical NuRD complex or in other Class I HDAC complexes, and only a subset are in complex with PWWP2A and recruited to gene bodies (see discussion). Furthermore chemical cross-linking may preferentially capture MTA2 and HDAC1/2 at promoters if they are more abundant or more stably associated with chromatin at promoters than at gene bodies.

**PWWP2A/B DKO lead to H3K9ac and H3K27ac gain at active genes.** To directly test role of PWWP2A/B and its associated HDAC complex in vivo, we established a genetic deletion in

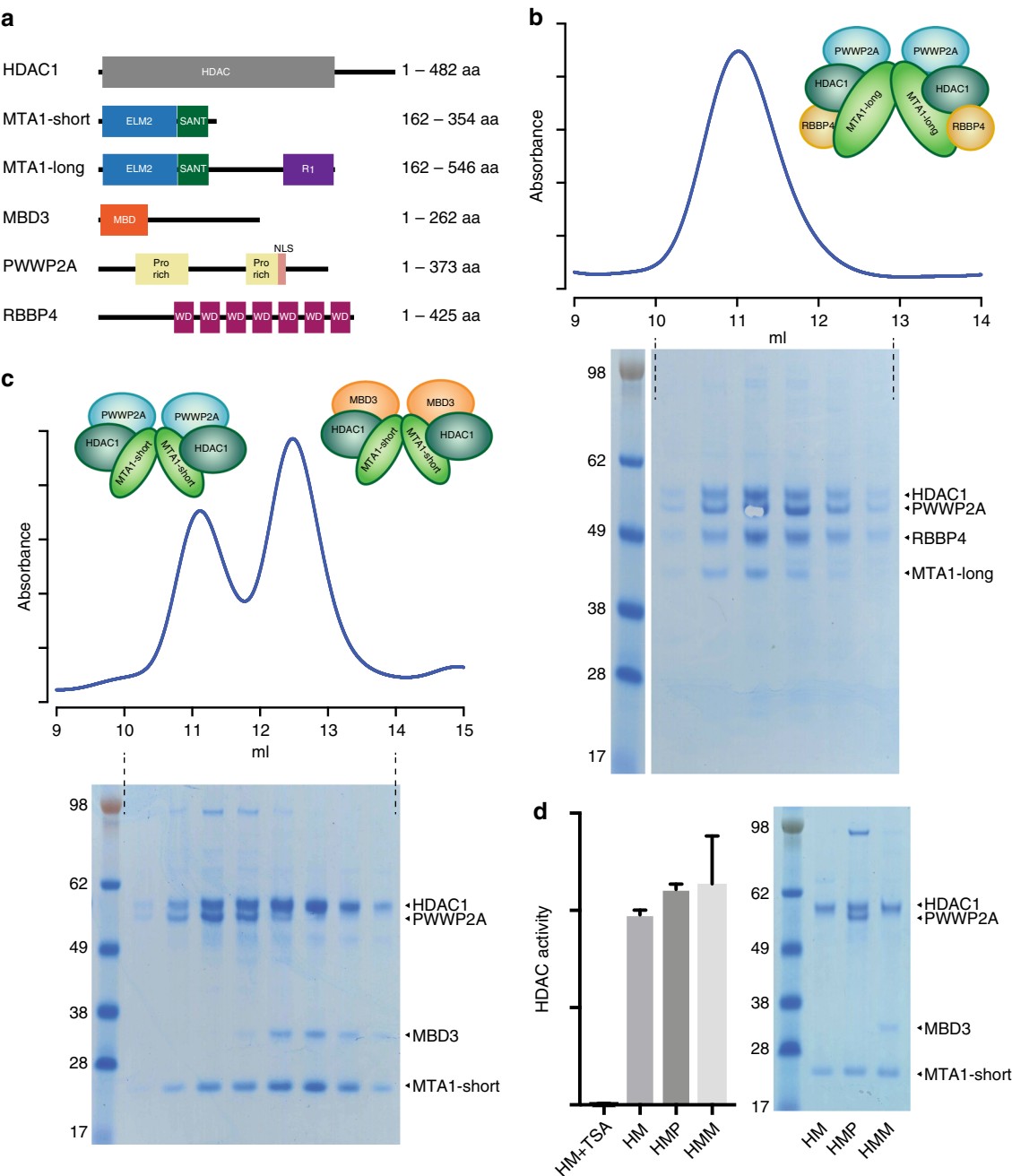

**Fig. 3** PWWP2A and MBD3 form mutually exclusive HDAC complexes with equivalent deacetylase activity. **a** Domain structures of NuRD complex components HDAC1, MTA1, MBD3, PWWP2A, and RBBP4. The MTA1-short construct recruits HDAC1 through the ELM2 and SANT domains and the extended MTA1-long construct contains an additional domain that recruits RBBP4 (as described in Millard et al.[19]). **b** PWWP2A is recruited to the larger core NuRD complex as shown by co-expressing with MTA1-long, HDAC1 and RBBP4. The identity of PWWP2A was confirmed by mass spectrometry following a gel stab of the protein band. **c** PWWP2A and MBD3 overexpressed with MTA1-short and HDAC1 in HEK293F cells elute as two mutually exclusive complexes as shown by gel filtration. The intensity of the PWWP2A band suggests that it is a stoichiometric component of the complex. MBD3 appears to be substoichiometric although this protein stains weakly with coomassie blue. **d** Histone deacetylation assay comparing the HDAC activity of complexes containing HDAC1 and MTA1-short (HM), containing HDAC1, MTA1-short and PWWP2A (HMP) and containing HDAC1, MTA1-short and MBD3 (HMM). TSA was used as a control to inhibit HDAC activity. Equivalent amounts of HDAC1 protein was used in each assay as judged by SDS-PAGE. The error bars denote SEM from three replicates

E14 mESCs. We generated double as opposed to single knockouts of PWWP2A/B due to our mass spectrometry data which suggest that PWWP2A and PWWP2B can co-exist in the same complex, and thus have overlapping functions (Supplementary Fig. 2b,d). Two independently derived PWWP2A/B DKO clones were generated using CRISPR/Cas9 to delete the majority of the coding region of both genes (Supplementary

Fig. 5a,b). Importantly, exons coding for both the NuRD-interacting domain and the PWWP domain were deleted. PWWP2A/B DKO ES cells retained ES morphology, pluripotency and the ability to undergo differentiation to embryoid bodies upon LIF withdrawal (Supplementary Fig. 5c). Global levels of NuRD complex components in these cells were not affected (Supplementary Fig. 5d).

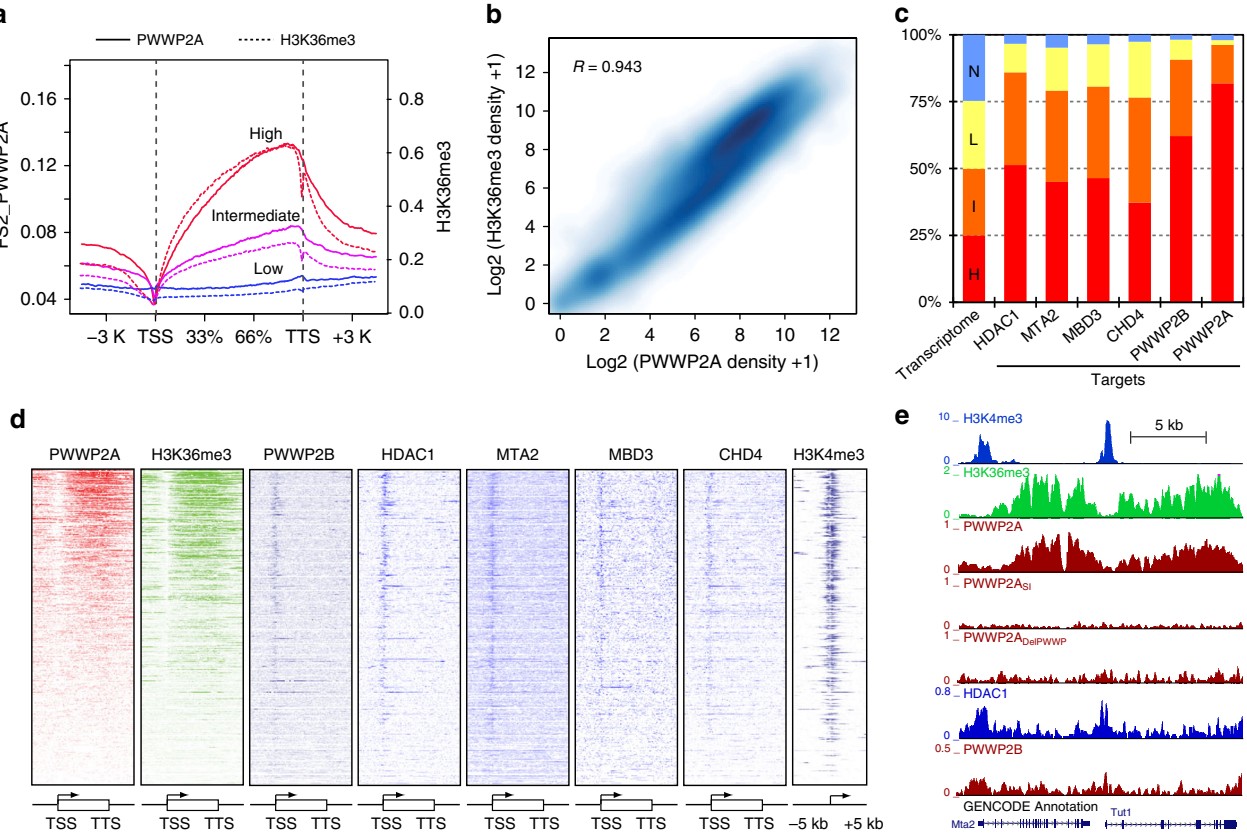

**Fig. 4** PWWP2A and PWWP2B are enriched at actively transcribed genes. **a** PWWP2A binding patterns are highly correlated with H3K36me3 at high, intermediate, and low expressed genes (from 5 kb upstream of TSS to 5 kb downstream of TTS). Solid line: PWWP2A; Dash line: H3K36me3. The x-axis denotes the normalized genomic regions and average coverage signals for each gene group, respectively. The left and right y-axis represent the signal of PWWP2A and H3K36me3, respectively. **b** Scatterplot showing the correlation of gene body occupancy by PWWP2A and H3K36me3. **c** Percentage of PWWP2A, PWWP2B, HDAC1, MTA2, MBD3, and CHD4 occupied genes by expression level (H High, I Intermediate, L Low, N No). The majority of PWWP2A and PWWP2B bound genes are highly expressed genes. **d** Heatmaps showing the binding profile across all gene loci for PWWP2A, H3K36me3, PWWP2B, HDAC1, MTA2, MBD3, CHD4 sorted by occupancy of PWWP2A from 5 kb upstream of the TSS to 5 kb downstream of the TTS. H3K4me3 is shown from −5 kb to +5 kb of the TSS. **e** A representative example of UCSC genome browser tracks displaying the ChIP-seq profile at the *Mta2-Tut1* locus. The long protein isoform of PWWP2A co-localises with H3K36me3. The short isoform PWWP2ASI and PWWP2AΔPWWP have no detectable gene body binding. The y-axis represents the normalized coverage

Calibrated ChIP-seq, a strategy used to quantitatively compare independent biological samples by normalising to spike-in *Drosophila* cells, was employed to study the effect of PWWP2A/B deletion on histone acetylation levels genome wide (Supplementary Fig. 5e and Methods). We performed calibrated ChIP-seq for H3K9ac and H3K27ac, two modifications enriched at active genes. In cells lacking PWWP2A/B, small increases in H3K9ac can be observed over the TSS flanking region (Fig. 5a). This increase in H3K9ac upon PWWP2A/B loss is greater at high occupancy targets (Fig. 5b), which are also genes with high expression level (Supplementary Fig. 5f). Increased levels of H3K27ac were also observed in the PWWP2A/B DKO cells (Fig. 5c). Similarly levels of H3K27ac were more perturbed at PWWP2A high occupancy targets (Fig. 5d) and genes with high levels of expression (Supplementary Fig. 5g). Small increases in H3K9ac across the gene body region were consistently observed, but increases in H3K27ac across the gene body was only seen for one of the two DKO clones (Supplementary Fig. 5h-k). While PWWP2A has the ability to associate with gene body chromatin, the role of this protein in mediating gene body deacetylation requires further analysis. These results show that PWWP2A/B contributes to regulating the levels of histone acetylation of all transcribed genes, with highly expressed genes being most targeted.

We also performed ChIP-seq of HDAC1 and MTA2 and found no significant changes in the binding of these proteins at PWWP2A high occupancy targets upon loss of PWWP2A/B (Supplementary Fig. 5l,m). We speculate that since ChIP for chromatin binding proteins is far less efficient than for histone modifications, it may not be sensitive enough to capture small changes in complex recruitment.

**PWWP2A/B DKO leads to transcriptional elongation defects**. We next sought to investigate how changes in histone acetylation at active genes affects transcription. We examined the effect of PWWP2A/B loss on both Pol II binding and nascent transcription. ChIP-seq of total RNA Pol II (antibody to the NTD) revealed that both DKO clones display a relative increase in promoter-proximal RNA Pol II binding compared to WT cells (Fig. 5e) which is suggestive of increased RNA Pol II promoter-proximal pausing upon PWWP2A/B loss. RNA Pol II pausing is a defining feature of almost all transcribed genes and is an important step in transcriptional regulation. Interestingly, increased Pol II pausing has also been observed in cells treated with Class I HDAC inhibitors SAHA and TSA[12,13]. The Pausing Index is a measure of the ratio of paused Pol II relative to elongating Pol II for each gene,

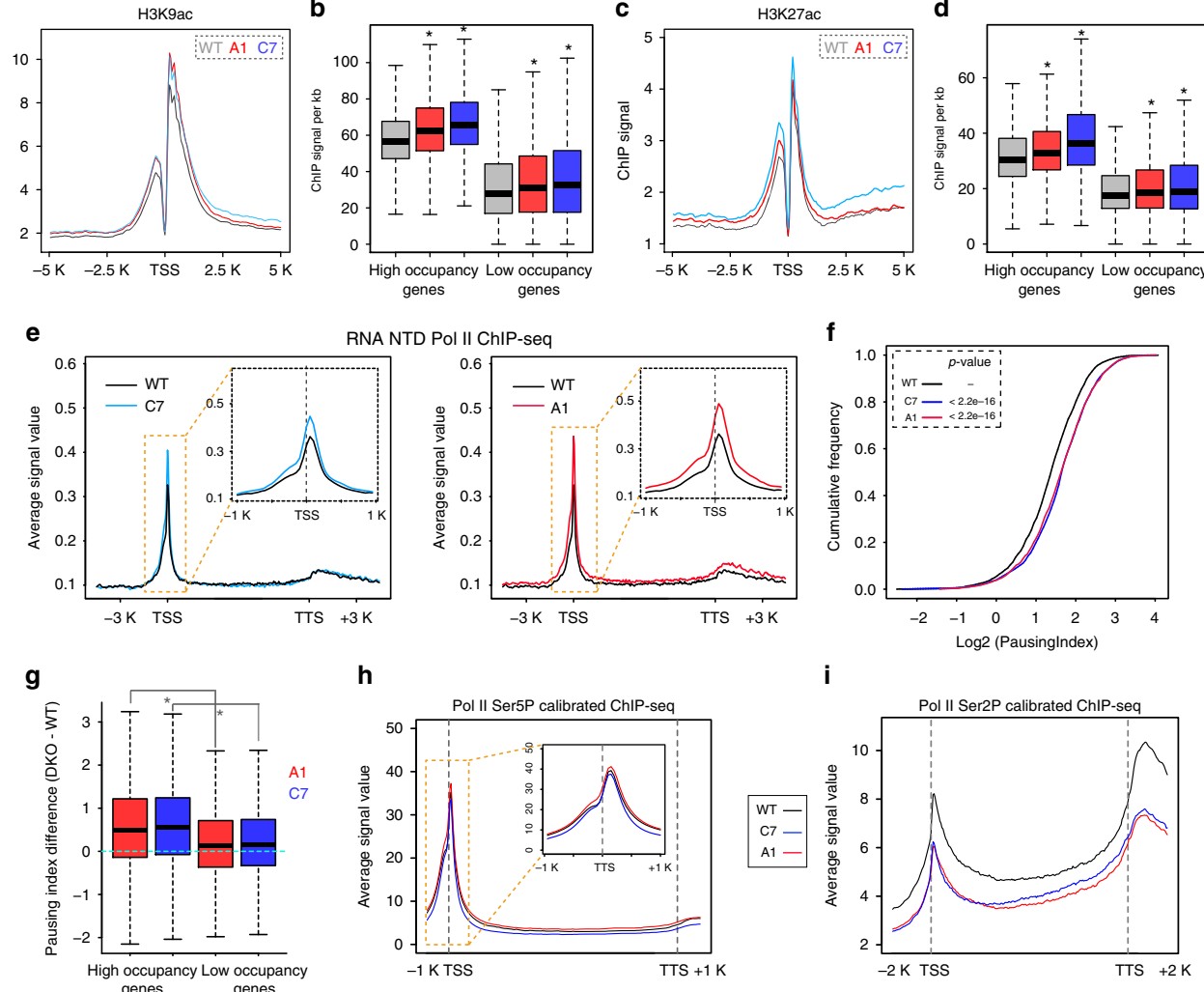

**Fig. 5** Loss of PWWP2A/B leads to an increase in H3K9ac and H3K27ac and Pol II elongation defect at PWWP2A occupancy genes. **a** Calibrated Meta-gene profile for H3K9ac (average of two biological replicates) at PWWP2A high occupancy genes in WT mESCs and two double knockout cells over the TSS ±5 kb region. **b** Boxplot showing the average calibrated H3K9ac ChIP-seq signal from two replicates at PWWP2A high and low occupancy genes over the region TSS ±5 kb. * indicates significance by Wilcoxon test. **c, d** same as (**a, b**) for H3K27ac calibrated ChIP-seq. **e** The meta-gene profile for RNA Pol II NTD ChIP-seq (average of two biological replicates) across the normalized genic regions and 5 kb flanking regions of PWWP2A high occupancy genes for WT (black) and two PWWP2A/B double knockout cells C7 (blue) and A1 (red). The y-axis represents the average signal. The inset figures show the signal around the TSS ±1 kb region. **f** The cumulative frequency plot showing the pausing index (based on RNA Pol II ChIP-seq) distribution for PWWP2A high occupancy genes in WT mESCs (black), and PWWP2A/B double knockout cells C7 (blue) and A1 (red). The p-value was calculated based on the one-sided Wilcoxon test. **g** Boxplot of the difference in the Pausing Index of PWWP2A/B knockout cells C7 (blue) and A1 (red) compared to WT at PWWP2A high occupancy and low occupancy genes. The asterisk indicates significance by Wilcoxon test. **h** The meta-gene profile for calibrated RNA Pol II Ser5P ChIP-seq (average of two biological replicates) across the normalized genic regions and 1 kb flanking regions of PWWP2A high occupancy genes for WT (black) and two PWWP2A/B double knockout cells C7 (blue) and A1 (red). The y-axis represents the average signal. The inset figure shows the signal over the TSS ±1 kb region. **i** The meta-gene profile for calibrated RNA Pol II Ser2P ChIP-seq (average of three biological replicates) across the normalized genic regions and 2 kb flanking regions of PWWP2A high occupancy genes for WT (black) and two PWWP2A/B double knockout cells C7 (blue) and A1 (red). For all the boxplots, the lower and upper edge of the box represents the first and third quartile, respectively. The horizontal line inside the box indicates the median. Whiskers identify the farthest data points within 1.5× the interquartile range (IQR)

calculated as the ratio of the read density of promoter-proximal Pol II to the read density of gene body Pol II (Supplementary Fig. 6a and Methods). We found that the Pausing Index is significantly increased in PWWP2A/B DKO cells compared with WT (Fig. 5f). In two biological replicates of both DKO clones, we observe a greater and statistically significant increase in RNA Pol II pausing at PWWP2A high occupancy genes compared with low occupancy genes (Fig. 5g). Similarly, increases in Pol II pausing upon PWWP2A/B deletion correlate with gene expression, and are increased to a greater degree at highly expressed genes

(Supplementary Fig. 6b), where PWWP2A/B is most enriched. PWWP2A/B deletion in mESCs has a small effect on nascent transcription as measured by 4sU RNA-seq with 90 genes displaying significant upregulation while slightly larger group of 306 genes are significantly downregulated (adj_p < 0.05; Supplementary Fig. 6c). The profile of nascent transcription at PWWP2A high occupancy genes is also similar to the effect in Pol II pausing. There is a relative increase in TSS-proximal reads in the sense and antisense direction compared with reads from the gene body region in the PWWP2A/B DKO cells (Supplementary Fig. 6d,e).

To better understand how Pol II dynamics is altered, we examined the profile of Pol II CTD phosphorylation in WT and DKO cells. Calibrated ChIP-seq was performed for Pol II Ser5P and Pol II Ser2P, phosphorylation states which are associated with paused and elongating Polymerase, respectively. The profile of Pol II Ser5P shows minor to no changes in PWWP2A/B DKO (Fig. 5h), leading us to speculate there is little disruption of Polymerase initiation. Levels of Pol II Ser2P on the other hand were decreased in the DKO cells particularly at PWWP2A high occupancy targets (Fig. 5i and Supplementary Fig. 6f). Global levels of total Pol II, Pol II Ser5P, and Pol II Ser2P were not significantly altered in DKO cells (Supplementary Fig. 6g). Thus it appears that the increase in the Pol II pausing upon PWWP2A/B deletion is due to defects in pause release or elongation.

Our observations of the effect of PWWP2A/B deletion on Pol II pausing are similar to what has been previously observed in studies using HDAC inhibitors, and is further evidence that HDAC activity at active genes may be required for the regulation of Pol II dynamics at the transition between Pol II pausing and elongation.

## Discussion

In eukaryotes multiple pathways for the recruitment of the NuRD complex to chromatin have been described. First, HDACs exhibit untargeted low-level binding genome-wide through the inherent affinity of HDACs for histone tails and nucleosomes[28]. Association of NuRD with chromatin-interacting proteins mediates its recruitment to regions with varying epigenetic modifications—including to methylated or hemi-methylated DNA by MBD2 and MBD3, respectively, to H3K9me3 enriched pericentric heterochromatin by BEND3[25], and to promoter regions by WDR5[29] and UpSET[30]. Finally, transcription factors as such Snail[31], Ikaros[32], TIF1β[26], Rb[33] have been shown to recruit NuRD to regulate a subset of genes in response to signalling pathways or in cell-fate decisions. Together, these mechanisms spatially and temporally regulate chromatin acetylation levels genome-wide.

In this study, we identify a family of proteins PWWP2A/B which are highly conserved in evolution and define a class of Class I HDAC complexes which are NuRD-like in composition. PWWP2A is the first protein characterised which binds to the deacetylase subcomplex of NuRD, and inhibits the association of canonical NuRD subunits MBD2/3, CHD3/4, and p66α/β. The mutual exclusivity between PWWP2A/B and MBD2/3 is consistent with published datasets denoting PWWP2A as co-purifying with MTA1[34] and HDAC1/2[21] but never with MBD3[35]. Quantitative mass spectrometry analysis of MTA1 interactors find that in HeLa cells 32% of immunoprecipitated MTA1 associates with MBD3, 28% with MBD2, and 19% with PWWP2A[34]. We find that amino acids 148–373 in PWWP2A is sufficient to mediate the interaction with MTA1:HDAC1, and a short motif (R/QPRQV/L) in this region conserved between MBD2/3 and PWWP2A/B can bind the MTA1:HDAC1 complex. We surmise that the association of PWWP2A with the deacetylase core of NuRD may inhibit formation of the holo-NuRD complex and contributes to the variation of NuRD and NuRD-like complexes that exist in vivo (Fig. 6).

PWWP2A/B and MBD2/3 confer different modes of chromatin recruitment to the MTA1/2/3-HDAC1/2 complex. MBD2 binds methylated DNA and is enriched at pericentric heterochromatin[36], while MBD3 localises to the enhancers and promoters of active and bivalent Polycomb target genes[37,38]. PWWP2A and PWWP2B on the other hand are enriched at highly active genes in mESCs, with PWWP2A predominantly at H3K36me3-rich gene bodies and PWWP2B at active promoters and enhancers. We find that the association of PWWP2A with

chromatin is highly dependent on its PWWP domain, as PWWP2A$_{SI}$ and PWWP2A$_{\Delta PWWP}$, which both lack the C-terminal PWWP domain, show a significant loss of chromatin binding. Uniquely, PWWP2A recruitment through recognition of H3K36me3 provides a close parallel with Eaf3 mediated HDAC recruitment in *Saccharomyces cerevisae*. In yeast, gene body deacetylation is critical for transcriptional fidelity by inhibiting cryptic initiation from within the coding regions of genes[3–5]. Whether PWWP2A/B loss is involved in the inhibition of spurious transcription initiation remains to be tested. PWWP2B does not appear to be enriched at H3K36me3-containing chromatin, suggesting that other mechanisms may regulate its targeting to chromatin. A recent study identified an H2A.Z binding domain in the internal stretch of PWWP2A (amino acids 292–574)[39]. H2A.Z is a histone variant associated with regulatory regions and is enriched at active promoters genome-wide[40,41]. By sequence homology, the H2A.Z binding domain appears to be conserved in PWWP2B and is partially present in the short isoforms of both paralogs. While the localisation of PWWP2A is dependent on the PWWP domain, H2A.Z binding may contribute more to PWWP2B's localisation on chromatin. The relative contributions of H2A.Z and H3K36me3-binding in the chromatin recruitment of PWWP2A and PWWP2B and their short isoforms requires further study. Due to the presence of two PWWP2A/B molecules in one deacetylase complex, we surmise that PWWP2A/B hetero and homo complexes may have both overlapping and paralog-specific functions, and that complex composition dictates the chromatin environments which can be sampled. Future experiments with single PWWP2A and PWWP2B KOs will be able to address the overlapping and paralog-specific functions of this family of proteins.

Furthermore, we were interested in understanding the biological role of this complex in vivo. PWWP2A and PWWP2B are expressed early in development and are highly enriched in brain tissues and PWWP2A has been shown to be critical in *Xenopus* head differentiation, and eye and brain development[39]. Biochemical reconstitution of the PWWP2A:MTA:HDAC complex reveals PWWP2A to be a stoichiometric member in this HDAC complex that shows equivalent deacetylase activity to the MBD3:MTA:HDAC complex in vitro. In mESCs lacking PWWP2A and PWWP2B, we observe a small but reproducible gain in H3K27ac and H3K9ac across the entire genic region. This effect is most prominent at genes highly bound by PWWP2A, which are also predominantly highly transcribed genes. Only these two histone modifications were tested, but it is possible that PWWP2A/B loss may perturb histone deacetylation of other lysine residues as well. Histones associated with active chromatin undergo dynamic acetylation turnover in multiple higher eukaryotes suggesting HDAC function at active chromatin is evolutionarily conserved[27]. Histone acetylation has been shown to play a role in gene expression through generating a chromatin environment favourable to transcription through mediating chromatin accessibility or histone exchange[42–45], regulating the various stages of the transcription cycle[45–47], or serving as transcriptional memory once the gene expression state is established[48–50]. Experiments with HDAC inhibitors show that loss of HDAC activity leads to global increases in acetylation at active genes, and increases in Pol II pausing and blockade in Pol II transcriptional elongation measured by GRO-seq[12]. We also observe that RNA Pol II dynamics are altered upon PWWP2A/B deletion. While levels of Ser5 phosphorylated Polymerase show no significant changes, levels of Ser2 phosphorylated elongating Pol II is decreased in cells lacking PWWP2A/B, suggesting a role for PWWP2A/B in promoting transcriptional elongation. Further work is need to address how changes in histone acetylation affect the

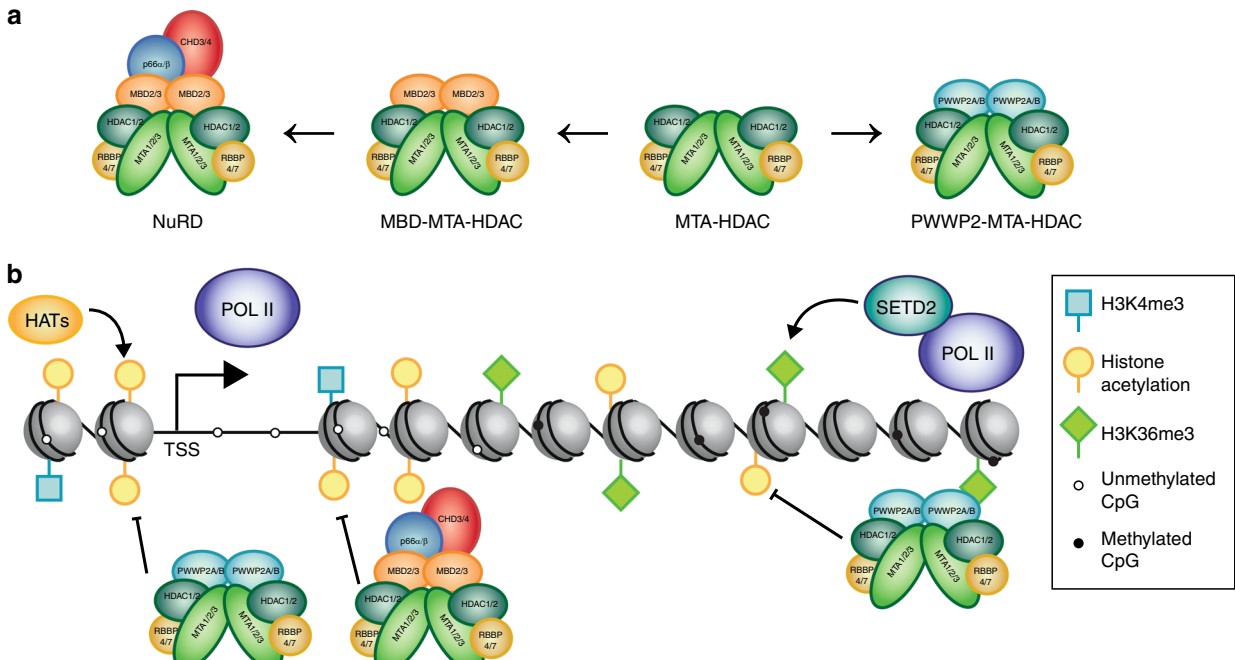

**Fig. 6** Proposed model for the assembly and recruitment of the variant PWWP2A/B-NuRD complexes. **a** The NuRD-family of HDAC complexes assemble around the MTA1/2/3:HDAC1/2:RBBP4/7 subcomplex. PWWP2A/B and MBD2/3 bind to the deacetylase subcomplex in a mutually exclusive manner. Canonical NuRD components p66α/β and CHD3/4 can only associate with the MBD2/3 containing complex. **b** SETD2 deposits H3K36me3 over gene body regions co-transcriptionally. PWWP2A/B recruits HDAC activity to the gene bodies of highly expressed genes through binding to H3K36me3. Both canonical NuRD and the PWWP2A/B HDAC complexes are recruited to regulatory promoter regions, but their activity is antagonised by HAT complexes which are also enriched at promoter regions. The regulation of histone acetylation at active genes facilitates RNA Pol II transcriptional elongation

epigenetic landscape at active genes and the recruitment of factors important in promoting Pol II pause release or elongation.

In this study, we identify a variant NuRD complex which is formed by association of PWWP2A/B with the MTA1/2/3: HDAC1/2 deacetylase subcomplex. Our work supports earlier studies which show that NuRD can assemble in a modular fashion from preformed subcomplexes[20]. This PWWP2A/B: MTA1/2/3:HDAC1/2 complex is targeted to actively transcribed genes genome-wide, where HDAC mediated deacetylation is important for the fine-tuning of RNA Pol II dynamics.

## Methods

**Histone purification.** (adapted from Dyer et al.[51]): The plasmids for the expression of canonical *Xenopus laevis* histones H2A, H2B, H3, H4 were obtained from the group of Timothy Richmond. The pET-histone expression plasmids were expressed in BL21 DE3 pLysS *E. coli* cells, grown in 2XTY medium and induced at 0.5 mM IPTG. Histones were purified from inclusion bodies under denaturing conditions (7 M urea, 20 mM sodium acetate, pH 5.2, 1 M NaCl, 5 mM 2-mercaptoethanol, 1 mM EDTA), over a HiPrep 26/60 Sephacryl S-200 HR column. Finally the purified histones were gradually refolded by dialyses against 3 × 4 L of water with 2 mM 2-mercaptoethanol at 4 °C.

**Generation of site-specific methyllysine analogs.** This was performed as in Simon et al.[23]. Briefly, to introduce site-specific methyllysine analogs into H3, the lysine of interest H3 K36 was mutated to a cysteine, and the only native cysteine in H3 at residue 110 is mutated to an alanine. The mutant H3 K36C C110A was expressed and purified as described above and alkylated under reducing conditions with (2-bromoethyl) trimethylammonium bromide to produce a trimethyllysine (Kme3) mimic. The reaction was monitored by mass spectrometry, the efficiency of the reaction was usually near completion with total conversion of the reactive cysteine to the trimethyllysine analog.

**Histone octamer assembly and purification.** A total of 3 mg of each histone H2A, H2B, H3 (unmodified or modified), H4 was unfolded in denaturing buffer (7 M guanidine HCl, 20 mM Tris, pH 7.5, 10 mM DTT), then mixed together at equal

ratios. The histones were then dialysed into 3 × 4 L refolding buffer containing high salt (2 M NaCl, 10 mM Tris, pH 7.5, 1 mM EDTA, 5 mM 2-mercaptoethanol). The dialysed sample was run over a HiLoad Superdex 200 16/60 column to separate the histone octamer from H3/H4 tetramers, H2A/H2B dimers, and individual histones. Octamers were stored at 4 °C for short term use, or in 50% glycerol at −20 °C for long-term storage.

**DNA purification and nucleosome reconstitution.** The plasmid containing the 2 × 601 Widom nucleosome positioning sequence was obtained from the lab of Jonathan Widom. The 2 × 601 sequence for nucleosome reconstitution was PCR amplified from the plasmid pBlu2SKP 2 × 601 + 48, which contains two 601 nucleosome positioning sequences separated by a 48 bp linker. The PCR product was purified by anion exchange in a 1 ml Resource Q column, ethanol precipitated, and finally dissolved into milliQ water.

Octamers and 2 × 601 DNA were mixed at 2:1 molar ratio to obtain dinucleosomes at high salt (2 M NaCl), and step wise dialysed into low salt buffer (100 mM NaCl, 10 mM Tris, pH 7.5). Nucleosome reconstitution was visualised by running the free DNA and nucleosomes in a non-denaturing 0.8% TB gel, then staining with EtBr.

**SILAC labelling of HeLa cells.** HeLa cells were cultured in RPMI medium minus arginine and lysine supplemented with 10% dialysed foetal calf serum, and with 76 μg/ml arginine and 133 μg/ml lysine (either light R0K0 or heavy R10K8). Trypsin-free cell dissociation buffer from invitrogen was used for passaging cells. Cells were passaged nine times to ensure full incorporation of heavy amino acids which was checked by mass spectrometry to be >98% heavy label incorporation.

**SILAC nucleosome affinity pull-downs.** (adapted from Bartke et al.[22]): Biotiny-lated nucleosomes were prepared fresh and checked for purify and correct histone stoichiometry. A total of 20 μg of nucleosomes was conjugated to 75 μl Pierce streptavidin magnetic beads for 1 h, then incubated for 4 h with 0.5 mg of nuclear extract pretreated with benzonase. For the forward experiment, unmodified nucleosomes were incubated with light extract, and H3Kc36me3 nucleosomes with heavy extract. In the reverse experiment, unmodified nucleosomes were incubated with heavy extract and H3Kc36me3 nuclesomes with light extract. After incuba-tion, samples were washed five times in the binding buffer 20 mM HEPES, pH 7.5, 150 mM NaCl, 0.2 mM EDTA, 20% glycerol, 0.1% NP-40, 1 mM DTT, and com-plete protease inhibitors, then boiled in SDS-PAGE loading buffer.

Samples were separated by SDS-PAGE to check the efficiency and quality of the pulldown. The paired samples were then mixed together for in solution trypsin digestion as described below and subsequent analysis by LC–MS/MS.

**SILAC mass spectrometry**. SILAC labelled protein digests were analysed on a LC–MS/MS platform consisting of Dionex Ultimate 3000 nUPLC and Q-Exactive mass spectrometer (both Thermo Fisher Scientific). The samples were separated with an Acetonitrile gradient of 2–35% in 0.1% formic acid and 5% DMSO over 60 min on an Easyspray column (75 μm × 500 mm, 250 nl/min flow). MS data were acquired on two LC–MS/MS plattforms (Orbitrap Fusion Lumos and Q-Exactive) with parameters outlined in Supplementary Data 6 (upper panel). Data was analysed with Maxquant[52] 1.5.2.8, Mascot 2.5 or PEAKS 8.5 using parameters as shown in the lower panel of Supplementary Data 6. The mass spectrometry proteomics data have been deposited to the ProteomeXchange Consortium via the PRIDE[53] partner repository with the dataset identifier PXD010445. Processed data can be found in Fig. 1 and Supplementary Data 1.

**Preparation of nuclear extracts**. The cells were collected by using either trypsin or scraping. The pellet was resuspended in hypotonic buffer (10 mM HEPES, pH 7.9, 1.5 mM MgCl$_2$, 10 mM KCl, 0.5 mM DTT, 0.5 mM PMSF, complete protease inhibitors). Lysis of the cytoplasmic membrane was achieved using hypotonic buffer containing 0.1% NP-40. Nuclei were washed, pelleted, and extracted for 1 h on ice in high salt buffer (5 mM HEPES, pH 7.9, 26% glycerol, 1.5 mM MgCl$_2$, 0.2 mM EDTA, 350 mM NaCl). The nuclei were centrifuged at maximum speed at 4 °C, and the supernatant was taken as the nuclear extract, snap frozen and stored at −80 °C.

**Flag immunoprecipitation**. Flag affinity purifications for co-IP experiments were carried out using the anti-Flag M2 affinity gel (Sigma-Aldrich). Nuclear extract was treated with Benzonase (Sigma-Aldrich) prior to use to eliminate DNA-facilitated protein–protein interactions. A total of 0.5 mg of nuclear extract was incubated with 50 μl of beads for 3 h at 4 °C. This was followed by 8–10 stringent washes in buffer containing (25 mM Tris, pH 7.5, 300 mM NaCl, 0.2% NP-40, complete protease inhibitors). Samples were boiled in SDS-PAGE loading buffer for downstream analysis by western blot. Uncropped westerns for all IPs can be found in Supplementary Figure 7.

**Fractionation of nuclear extract by gel filtration**. To determine the co-migration of PWWP2A with NuRD components, nuclear extract was prepared from the E14 cell line stably expressing FS2-PWWP2A. A quantity of 0.3 mg nuclear extract was treated with Benzonase for 1 h, centrifuged at maximum speed for 10 min at 4 °C, then loaded on a Superose 6 10/300 column equilibrated with 20 mM Tris, pH 7.5, 300 mM NaCl, 2 mM MgCl$_2$, 5% glycerol, 1 mM PMSF, complete protease inhibitors. A volume of 0.25 ml fractions was collected and precipitated with TCA. The protein pellets were resuspended and boiled in SDS-PAGE loading buffer, and every other fraction was loaded for western blot analysis. Uncropped westerns for this fractionation can be found in Supplementary Figure 7.

**Streptactin purification**. For each mass spectrometry experiment, cells were collected from 15 cm diameter tissue culture dishes (×40) corresponding to 2–3 × 10$^9$ cells. Nuclear extract was prepared in advance and treated with Benzonase for 1 h just prior to use. A quantity of 15 mg of nuclear extract was incubated with 50 μl of streptactin agarose beads for 4 h. The samples were then washed for 8 × 15 min in high salt wash buffer (20 mM Tris, pH 8.0, 500 mM NaCl, 0.2% NP-40, 5% glycerol, 1 mM DTT, complete protease inhibitors), and finally eluted with 10 mM desthiobiotin in 20 mM Tris, pH 8.0, 150 mM NaCl, 0.2% NP-40, 5% glycerol, 1 mM DTT, and protease inhibitors (Complete ®). Samples were snap frozen and stored at −80 °C. Raw data can be found in Supplementary Data 2–4.

**Preparation of samples for mass spectrometry**. IP samples were checked for quality by silverstain and western blot prior to processing for mass spectrometry. The sample was reduced with 5 mM DTT for 30 min and alkylated with 20 mM iodoacetamide for 30 min both at room temperature. Proteins were precipitated by methanol/chloroform extraction[54], and the pellet was solubilised in 6 M urea buffer. In-solution digestion was carried out using mass spec grade trypsin (Promega) overnight at 37 °C at a ratio of 1:50. The digestion was stopped by the addition of formic acid (1% final concentration), and the peptides were purified using SEPPAK C18 cartridges. Briefly, the SEP-PAK C18 column was eliquibrated with 5 ml of buffer B (65% acetonitrile, 35% milliQ water, 0.1% TFA), followed by 10 ml Buffer A (98% milliQ water, 2% acetonitrile, 0.1% TFA). Bound peptides were washed with 10 ml buffer A and eluted in buffer B. Samples were dried under vacuum and stored at −80 °C until analysis by LC–MS/MS.

**Protein expression in HEK293F cells**. HDAC complexes were expressed and purified from HEK293F cells as described in ref. [18]. Briefly, His$_{10}$-Flag$_3$-TEV-MTA1 was co-transfected with untagged versions of HDAC1, PWWP2A, MBD3, and RBBP4 in suspension-grown HEK293F cells (Invitrogen) with the transfection reagent polyethylenimine (PEI; Sigma). Cells were collected after 48 h, lysed by

sonication, and protein complexes were purified on FLAG resin (Sigma). Following TEV cleavage from the resin, gel filtration chromatography was performed on a Superdex S200 column (GE Healthcare) in buffer containing 25 mM Tris-Cl (pH 7.5), 50 mM potassium acetate, and 0.5 mM TCEP.

**HDAC activity assays**. HDAC1 activity was measured using a BOC-Lys-AMC substrate. Protein complexes containing equivalent amounts of HDAC1 (as judged by SDS-PAGE) were incubated with 100 μM BOC-Lys-AMC in a final volume of 50 μM buffer containing 50 mM Tris, pH 7.5, 50 mM NaCl, 5% Glycerol. The assay was developed by adding 50 μl of developer solution (50 mM Tris, pH 7.5, 100 mM NaCl, 10 mg/ml trypsin, 2 mM TSA). Fluorescence was measured at 335/460 nm using a Victor X5 plate reader (Perkin Elmer).

**Fluorescence anisotropy**. Fluorescence anisotropy of fluorescein-labelled PWWP2A peptide and unlabelled HDAC1:MTA1$_{162–354}$ complex was recorded in a black 96-well plate (Corning) at room temperature using a Victor X5 plate reader (Perkin Elmer). The PWWP2A peptide (residues 311–325) was amino-terminal labelled with 5-carboxyfluorescein (5-FAM) (Cambridge Research Biochemicals, Billingham, UK). Measurements were conducted in buffer containing 50 mM Tris, pH 7.5, 150 mM KAc, 0.15 mg/mL BSA, and 0.03% Tween-20. Fluorescence was excited at 480 nm and monitored at 535 nm.

**Cell culture**. E14 cells were grown in Dulbecco's Modified Eagle Medium (DMEM, from Life Technologies) supplemented with 10% foetal calf serum (FCS, from Seralab), 2 mM L-glutamine, 1X non-essential amino acids, 50 μM β-mercaptoethanol, 50 g/mL penicillin/streptomycin (all from invitrogen) and 1000 U/mL of LIF. Non-differentiating ES cells were grown on tissue culture dishes coated with PBS+1% gelatine. Cells were passaged using 0.05% trypsin-EDTA (Life Technologies) with 2% Chicken Serum (Life Technologies) and frozen in FCS +10% DMSO. For cells stably expressing PWWP2A/B or PWWP2A truncations, ES cell medium was supplemented with 1.75 μg/ml Puromycin.

SG4 Drosophila melanogaster cells were cultured at 27 °C with Schneider's Drosophila Medium (Thermo Fisher) supplemented with penicillin/streptomycin and 10% FCS.

**Generation of cell lines**. For the generation of PWWP2A/B DKO cells, sgRNAs targeting Pwwp2a and Pwwp2b were cloned into plasmid pSpCas9(BB)-2A-Puro (pX459). E14 cells were seeded on six-well plates and transfected with 2 μg of plasmid using Lipofectamine 2000 (Thermo Fisher). Cells were plated at different densities in three Petri dishes 24 h after transfection. A day after, medium was replaced by ES medium with Puromycin (2 μg/mL) for 48 h, after which cells were grown in ES medium until ES colonies were ready to be picked. Screening of knockout clones was achieved by PCR from genomic DNA, and were further characterized by Sanger sequencing of the PCR product.

The cDNA encoding full-length PWWP2A (and various truncations) and PWWP2B were cloned into pCAG-IRES-Puro mammalian expression plasmid with an N-terminal Flag-StrepII-StrepII tag, then transfected into E14 cells with Lipofectamine 2000. Stable integrants were select with 2 μg/mL Puromycin, and clones of varying expression levels were kept. The oligos and primers used in this study can be found in Supplementary Table 2.

**Chromatin immunoprecipitation**. For cross-linking-ChIP, 40–45 million ES cells were resuspended in 10 ml PBS with 2 mM EGS for 45 min, followed by 1% formaldehyde for a further 10 min. The reactions were quenched by the addition of glycine to a final concentration of 125 mM. Cells were first lysed for 10 min on ice in LB1 (50 mM HEPES, pH 7.9, 140 mM NaCl, 1 mM EDTA, 10% glycerol, 0.5% NP-40, 0.25% Triton). The nuclei were pelleted and washed for 5 min in LB2 (10 mM Tris, pH 8.0, 200 mM NaCl, 1 mM EDTA, 0.5 mM EGTA), and pelleted again. The nuclei were suspended in 1 ml of LB3 (10 mM Tris, pH 8.0, 100 mM NaCl, 1 mM EDTA, 0.5 mM EGTA, 0.1% sodium deoxycholate, 0.5% N-lauroylsarcosine) and sonicated using a BioRuptor sonicator (Diagenode) with for 25 cycles 30 s on and 30 s off, followed by centrifugation at maximum speed for 15 min at 4 °C, and either used freshly or snap frozen and stored at −80 °C. For calibrated ChIP-seq[55] of Pol II Ser5P and Pol II Ser2P, 40 million ES cells and 10 million Drosophila SG4 cells were pooled in PBS prior to cross-linking. For each IP, 100 μl of chromatin was diluted with ChIP dilution buffer (1% Triton X-100, 1 mM EDTA, 20 mM Tris-HCl, pH 8, 150 mM NaCl) prior to pre-clearing with protein A agarose beads or magnetic Dynabeads (Invitrogen) blocked with 1 mg/mL BSA and 1 mg/mL yeast tRNA at 4 °C for 1 h. Precleared chromatin samples were further incubated overnight with relevant antibodies at 4 °C with rotation. For each calibrated ChIP sample 5–8 μg antibody was used. Antibody-bound chromatin was isolated on blocked protein A beads, and followed by washes with low salt buffer (0.1% SDS, 1% Triton, 2 mM EDTA, 20 mM Tris-HCl (pH 8.1), 150 mM NaCl), high salt buffer ((0.1% SDS, 1% Triton, 2 mM EDTA, 20 mM Tris-HCl (pH 8.1), 500 mM NaCl), LiCl buffer (0.25 M LiCl, 1% NP-40, 1% sodium deoxycholate, 1 mM EDTA, 10 mM Tris-HCl (pH 8.1)), and twice with TE (10 mM Tris-HCl (pH 8.0), 1 mM EDTA) buffer. The DNA was eluted with 1% SDS and 100 mM NaHCO$_3$, and reversed cross-linked at 65 °C overnight in the presence of 200 mM NaCl, following by treatment with RNaseA and Proteinase K. DNA was purified by ChIP

DNA Clean and Concentrator kit (Zymo) and quantified by dsDNA Qubit reagents.

For calibrated native ChIP (H3K27ac and H3K9ac), 40 million ES cells and 10 million *Drosophila* SG4 cells were pooled and lysed in RSB (10 mM Tris, pH 8, 10 mM NaCl, 3 mM $MgCl_2$) + 0.1% NP-40 for 5 min on ice with gentle inversion. The nuclei were resuspended in 1 ml of RSB + 0.25 M sucrose + 3 mM $CaCl_2$, treated with 200U of MNase (Fermentas) for 5 min at 37 °C, quenched with 4 μl of 1 M EDTA, then centrifuged at $2500 \times g$ for 5 min. The supernatant was transferred to a fresh tube as fraction S1. The chromatin pellet was incubated for 1 h in 300 μl of nucleosome release buffer (10 mM Tris, pH 7.5, 10 mM NaCl, 0.2 mM EDTA), carefully passed five times through a 27 G needle using a 1 ml syringe, then centrifuged at $2500 \times g$ for 5 min. The supernatant from this S2 fraction was combined with S1. For each ChIP reaction, 80 μl of chromatin was diluted in Native ChIP incubation buffer (10 mM Tris, pH 7.5, 70 mM NaCl, 2 mM $MgCl_2$, 2 mM EDTA, 0.1% Triton) to 1 ml and incubated with antibody overnight at 4 °C. Samples were incubated with 50 μl protein A agarose beads preblocked in Native ChIP incubation buffer with 1 mg/ml BSA and 1 mg/ml yeast tRNA, then washed for a total of four times in Native ChIP wash buffer (20 mM Tris, pH 7.5, 2 mM EDTA, 125 mM NaCl, 0.1% Triton) and once in TE. The DNA was eluted with 1% SDS and 100 mM $NaHCO_3$, and was purified using ChIP DNA Clean and Concentrator kit (Zymo).

Approximately 5–20 ng of ChIPed DNA was used for library prep using NEBNext Ultra II DNA Library Prep Kit with NEBNext Single index, and then further quantified by qPCR with KAPA Library Quantification DNA standards (KAPA Biosystems) and SensiMix SYBR (Bioline, UK). The libraries were pooled and $2 \times 42$ paired-end sequencing was performed on the Illumina NextSeq500. The antibodies used for ChIP in this study can be found in Supplementary Table 1.

**ChIP-seq analysis**. ChIP-seq data for H3K36me3, H3K4me3, H3K27me3, H3K4me1, H3K27ac, H3K9ac, NANOG, OCT4, and CTCF were obtained from ENCODE Project (www.encodeproject.org/), CHD4 data from GSE64825, and MBD3 from GSM1246867.

The raw sequencing reads were mapped to the mm10 reference genome (conventional ChIP) or mm10 concatenated with dm6 genome (calibrated ChIP) using Bowtie[56]. PCR duplicates were removed by 'picard-tools MarkDuplicates' program, and only unique read pairs were kept for further analysis. MACS2 (2.0.10)[57] was employed to call the ChIP-seq peaks with the default parameters except (-f BAMPE -g mm –q 0.01). We defined the PWWP2A high occupancy genes (targets) by overlapping the called peaks with the annotated gene body regions, identifying 1963 genes (2955 transcripts). The HDAC1 targets were defined if the peaks were fully within the 10 kb regions surrounding the transcriptional start site (TSS) (i.e., 5985 genes in total). Targets of MBD3, CHD4, MTA2, and PWWP2B were called similarly to HDAC1. Gene Ontology enrichment analyses were performed for PWWP2A targets using the online tool DAVID[58]. The enrichments were measured by $-\log10$ (FDR-adjust $P$-value). The metagene binding patterns and heatmap matrix were generated by DANPOS[59] with the key parameters (–flank_up 5000 –flank_dn 5000 –region_size 10,000 –bin_size 50 –excludeP 0.01). The heatmaps were sorted by PWWP2A occupancy and plotted using a custom R script. The bigwig tracks shown in UCSC Genome Browser were generated using the custom script xbam2wig.py from https://github.com/guifengwei/glib, normalizing to one million mapped reads for PWWP2A and PWWP2B ChIP-seq, whereas 10 million mapped reads for the remaining calibrated ChIP-seq. The average of calibrated ChIP-seq signal was calculated from 10-million-library-size normalized values, shade area for Pol II Ser2P and Ser5P represents the standard error of mean by three biological replicates. The mapping summary and normalisation factors (library size) used in the ChIP-seq are listed in Supplementary Data 5.

**4sU-RNA immunoprecipitation**. 4sU-RNA-seq was performed as described in Rabani et al.[60]. In brief, cells were incubated with the medium supplemented with 500 μM 4sU for 15 min, after which medium was rapidly removed and cells were lysed with 5 mL of Trizol reagent (Life Technologies). Total RNA was treated with Ambion DNA-free DNase Treatment kit (Life Technologies) and resuspended in water. For each μg of total RNA, 2 μL of Biotin-HPDP (Pierce, 50 mg EZ-Link Biotin-HPDP) dissolved in DMF at a concentration of 1 mg/mL, and 1 μL of 10X Biotinylation buffer (100 mM Tris-HCl, pH 7.4, 10 mM EDTA), was added. After incubation for 15 min at 25 °C, the RNA was purified using two rounds of chloroform purification in Phase Lock Gel Heavy Tubes (Eppendorf), precipitation with an equal volume of isopropanol and 1/10 volume of 5 M NaCl, and finally resuspended in water.

Biotinylated 4sU-RNA was recovered using the μMacs Streptavidin Kit (Miltenyi). Per microgram of recovered biotinylated 4sU-RNA, 0.5 μL of streptavidin beads were added, in a total volume of 200 μL. Samples were washed six times, eluted in fresh 100 mM DTT, and further purified using the RNA Clean & Concentrator-5 kit (Zymoresearch). Libraries for RNA-seq were constructed using NEBNext® Ultra™ II Directional RNA Library Prep Kit for Illumina®.

**4sU and RNA-seq analysis**. The two replicates of RNA-seq data for the E14 mES cell line were retrieved from the mouse ENCODE project. The raw reads

from our 4sU data and ENCODE data were mapped to the mm10 reference genome with STAR[61] under the default parameters apart from (–outFilterMultimapNmax 1 –outFilterMismatchNmax 8 –alignEndsType EndToEnd). The gene expression level (FPKM) was called with Cufflinks[62]. We divided the genes equally into four groups (i.e., High, Intermediate, Low and No) based on expression level.

**ChromHMM analysis**. Using ENCODE ChIP-seq data for various histone modifications and transcription factors, we defined 12 candidate chromatin states (enhancer, promoter, gene body, CTCF-binding sites, etc) using ChromHMM (v1.11)[63] according to instructions (https://github.com/guifengwei/ChromHMM_mESC_mm10). We used the peaks called by MACS2 to calculate the enrichment of HDAC1, MTA2, PWWP2A, PWWP2B, and DNaseI across the 12 chromatin states.

**Polymerase pausing index analysis**. The pausing index is defined as the ratio of the reads density within promoter regions ($-200$ to 300 bp relative to transcription start site) over the reads density within the gene body (300bp downstream of the TSS $-1000$ bp downstream of transcription termination site). For genes with multiple isoforms, the isoform (longer than 200 nt) with the maximum pausing index was taken for further analysis.

**Statistical analysis**. Various statistical methods were used as appropriate and are indicated in the corresponding figure legends. The $P$-values of the ChIP-qPCR analysis were calculated on the basis of Student's $t$-test with $P < 0.05$ as significance level and shown as mean ± SEM. The two sided non-parametric Mann–Whitney test was performed to examine the significance in the Pausing Index analysis. For all the boxplots, the lower edge of the box represents the first quartile and the upper edge represents the third quartile. The horizontal line inside the box indicates the median. Whiskers identify the farthest data points within 1.5X the interquartile range (IQR). All the statistical analysis was performed in R.

## Data availability
All the high-throughput data (ChIP-seq, 4sU-RNA-seq) generated in this study have been deposited in GEO under GSE112114. Mass spectrometry data are available via ProteomeXchange with identifier PXD010445.

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

## Acknowledgements

We would like to thank all the members of the Brockdorff, Schwabe, and Klose labs for helpful suggestions and discussion, especially Emilia Dimitrova, Vincenzo di Cerbo, Hamish King, and Rob Klose, and Greta Pintacuda for critical reading of the manuscript. We would like to acknowledge Amanda Williams from Oxford Zoology sequencing for practical and technical assistance and Wei Li (Baylor College of Medicine) for computational analysis suggestions. T.Z. is supported by graduate scholarships from the Oxford Clarendon Fund, University College, and the Natural Sciences and Engineering Research Council of Canada (NSERC). Work in the Brockdorff lab is funded by grants from the Wellcome Trust (103768) and the European Research Council (340081). J.W.R.S. is supported by a Senior Investigator Award (WT100237) from the Wellcome Trust. J.W.R. S. is a Royal Society Wolfson Research Merit Award Holder. The Kessler lab was funded by grants from the Kennedy Trust, John Fell Fund 133/075, the Wellcome Trust (097813/Z/11/Z) and the Engineering and Physical Sciences Research Council (EP/N034295/1).

## Author contributions

T.Z. and N.B. conceived this study. T.Z. performed all the biochemical experiments, apart from those with the purified complexes which were designed and performed by C.J.M. and J.W.R.S. The NGS studies were designed by T.Z. and G.W. and all bioinformatics analysis was done by G.W. R.F., R.K., and B.M.K. provided advice and assistance in the design, processing, and analysis of the SILAC and MS experiments. T.Z. drafted the manuscript with input from all the authors.

## Additional information

**Competing interests:** The authors declare no competing interests.

