## [Peer Review File · Nature Communications]

Reviewers' comments:

Reviewer #1 (Remarks to the Author):

The manuscript by Zhang et al. describes the identification of a protein "reader" of the histone H3 mark, H3K36me3, the protein PWWP2A. The authors further characterize the complex PWWP2A is found in and determine that it interacts with HDAC1/2, MTA and RBBP4/7 (NuRD complex) and other proteins. The authors further demonstrate that the interactions are dependent on the PWWP domain of the PWWP2A protein. Lastly, the authors show that this protein is found on chromatin that contains H3K36me3, and also to some extent regulates the binding of some of the interacting proteins (i.e. HDAC1) to these same genes. Overall, the experiments are well put together and manuscript for the most part clearly written, and I recommend publication of this manuscript in Nature Communications,. Some questions I had were:

1. From the proteomics data in Figure 1, the enrichment of PWWP2A and other interactors such as HDAC2, RBBP7, and HTLF are quite weak, especially compared to MSH6 (the previously known H3K36me3 binder). Were statistical tests (i.e. t-tests or ANOVA) used to show that the enriched proteins were significant? Were these analyses only repeated twice?
2. Can the authors better explain why loss of PWWP domain of PWWP2A protein only affects enrichment of gene body versus promoter regions?
3. The ChIP-Seq data in Figure 4 is a little confusing and contradictory to the data presented in Figure 1E. Seems like the genome-wide patterns for H3K36me3, PWWP2A are in the body of the genes, but HDAC1, MTA2, MBD3 are most at the TSS.

Reviewer #2 (Remarks to the Author):

In this manuscript, Zhang and colleagues provide a new H3K36me3 interacting factor – PWWP2A/B – identified through a biochemical MS screen. This protein has interesting associations with known chromatin regulators that are probed in biochemical assays. Function in chromatin and gene regulation is pursued in knock-out cell lines using genomic techniques.

I find the first portion of this manuscript – the biochemistry of discovery – to be high quality, compelling and of high interest.

I find the second portion of this manuscript – the functional genomics analysis – to be problematic in many respects which I detail below. I urge the authors to consider the conclusions drawn from the data presented – I find many to be unsupported by the data.

Major questions:

1. I find it disturbing that MTA1/2/3 are not identified in the initial nucleosome screen. If the complex discussed in the text is relevant to K36me3, why is MTA1/2/3 not enriched? This leads one to wonder whether the biochemistry provided from 293F cells reflects the species detected by nucleosome binding. (Note that I find the biochemistry from 293F cells to be of high quality).

I would really like to see the 'data not shown' of PWWP2 lack of interaction with HDAC1 alone. I would also like to see PWWP2 interaction (or lack thereof) with HDAC1 coexpressed with RBBP7 (recalling that Schreiber's original description of HDAC1 was a 2 protein complex – HDAC with RBBP4/7).

The bottom line for me is that I am not certain that the species identified by nucleosome binding and the species identified biochemically in 293F cells (and by the other co-IP/western experiments) are the same thing. This is central to the story and conclusions. Please help me to understand.

2. While I certainly applaud the effort to produce DKO cells and do functional genomic experiments, I find those presented here to be hard to understand and certainly think the results

do not support the conclusions.

The data around pausing are very problematic for me. The authors show very small changes in the pausing index by metagene plot. More disturbing is the candidate gene shot in 6C. It looks like very marginal changes in pausing index in a highly expressed gene, with big changes in pausing index at a gene expressed at very low levels.

Is the global change indicated in 6a-c driven by large changes in pausing index at barely expressed genes?

The conclusions around changes in K9Ac are likewise problematic for me. I do not believe that the data in S4f support the authors conclusions that K9Ac increases at PWWP target genes.

Perhaps most importantly, I do not see in this data compelling evidence for colocalization in the genome of PWWP2A with HDAC1 or MTA2. The heat map in Figure 4D is not informative – the data for MTA2, HDAC1, MBD3, CHD4 are presented in a completely different manner than PWWP2A and K36me3. Figure 4e seems to me to conclusively demonstrate a lack of colocalization of PWWP2A with HDAC1 (which does colocalize with PWWP2B and K4me3 – but not with K36me3).

I do not understand how the authors reach the conclusion drawn in this section of the manuscript given the data presented.

Minor questions:

1. The recombinant nucleosome fishing experiment is very interesting and I like the approach. The validation by id of a known K36me3 binding factor is nice.

I wonder about proteins excluded from K36me3 – LUC7L1, TBL2, ... - can the authors provide any rationale why these proteins should be excluded from modified nucleosomes?

2. I find the gel filtration chromatography experiment less than compelling for clear demonstration of separate complexes. I suggest the authors explore a different matrix – perhaps ion exchange – that will give baseline resolution of PWWP and its pairing partners with conventional NuRD complex.

3. The co-IP experiments with FLAG (2d) would benefit from negative controls (i.e. PWWP 1-148, PWWP 373-649)

4. Methods for chip in ES cells could be more detailed. Reference to a previous paper requires the reader to search for how the experiment was performed. Please provide the methods here.

5. Please validate expression of MTA2 and HDAC1 in the DKO cells. Does loss of all PWWP2A/B result in decreased expression of these proteins?

6. Figure 5e is very difficult to follow. Please consider a different presentation.

Reviewer #3 (Remarks to the Author):

The authors use a quantitative proteomic approach to identify protein complexes that bind H3K36me3, leading to the discovery of a new variant of the NuRD complex. They verify the exclusive choice between components specific to the two variants (previously known one and the newly identified one) using orthogonal biochemical assays. They then use ChIP to identify the genomic location of the NuRD complex variants and use genetic perturbations to dissect their potential function.

Discussion:

This manuscript can be roughly divided to two parts; the first is a biochemical study of protein complexes associated with H3K36me3 and with their binders. The second is a genomic study of the impact of deletion of two of the binding proteins PWWP2A/B on chromatin marks and transcription. The first part provides clear and compelling evidence for the role of PWWP2A/B in NuRD complex formation and their potential function in chromatin binding. The second part is more complex and less clear. The analysis of localization of NuRD complex components and the effect of double KO of PWWP2A/B on chromatin marks identifies clear pattern which are consistent

with the biochemical results (binding preferences of PWWP2A/B and HDAC activity of NuRD). From these, however, it is hard to infer a mechanistic role of these NuRD variants in transcription. The PolII/4sU analysis is sketchy and less compelling (see comments below). Thus, it is hard to understand whether the changes in PolII pausing are due to missing deacetylation step in the DKO or due to various potential indirect effects (some of which are discussed in lines 354-362).

My recommendation is either to remove the last part and publish the solid aspect of the results, or to strengthen the results in this last part with proper analysis to understand whether the effects are due to changes in acetylation levels or to missing activity of the PWWP2-MTA-HDAC complex. Comparison to global HDAC inhibition will further define the specific contribution of the PWWP2 complexes in these effects.

Comments:

* Double KO (Fig 5) – why such discrepancies in the signal of the double KO ChIP?

* Changes in K9/27ac – clearly the regions of acetylation are of varying length. I am not sure if total ChIP signal is the right measure to look at the distribution. The dynamic range in the distribution among genes (targets/non-targets) is much larger than the differences between strains. If there are confounding factors (gene length, expression levels, etc) a paired analysis (ratio of change per gene between DKO and WT) will provide a tighter and more informative distribution.

* Analysis of PolII pausing is done in a brief and unconvincing manner. In the analysis of Fig4 the authors used spike-ins to normalize the signal (“calibrate ChIP” in their methods). They do not use similar calibration in the analysis of Pol II signal. The immediate question that comes to mind is whether there are significant changes in total amount of active/paused transcription in the DKO strains. Since NuRD, like most transcriptionally related machinery, is generic and has effects on all (or most) actively transcribed genes, one has to be careful in analysis of signal without proper study of absolute levels. For example, the global increase of acetylation levels might lead to sequestering of PolII at many abnormal sites, leading to reduction of PolII at bona-fide transcripts. Alternatively, cell might compensate and generate higher levels of PolII leading to accumulation of paused PolII at gene starts (traffic jams) while the actual levels of productive transcriptions remain more or less constant. Without calibration changes in the relative signal are hard to interpret, and various potential models (as the above examples) might provide “reasonable” account of the phenomena.

* 4sU-seq analysis – I really can’t tell what is shown in Supp Fig 4i and how it “shows a similar trend”

Minor issues:

Fig1c – I am curious why there is a substantial correlation between H/L in the two experiments? The non-specific signals seem as strong as the actual signal.

Fig 4c – I did not find this figure especially clear nor informative on top of the very clear signal we see in Figs 4a and b.

Line 238 “data now shown”?

Fig 5 – key of the color code on the figure itself will help the reader.

Fig 5e – the overlay is hard to read. In the top tracks the transparent red/blue are on top of the gray while on the bottom tracks they are obscured by the gray track. I appreciate that this is a lot of information to convey. The technical repeats are very consistent, which is great, but showing all

of them here (rather than a sup fig) creates information overload that takes from the main message of the figure.

Fig 5f – use consistent labels (e.g., E14 vs WT etc), preferably an informative one (DKO1 rather than C7)

We thank the reviewers for their thoughtful and helpful comments. Our responses are detailed below.

Reviewer #1 (Remarks to the Author):

The manuscript by Zhang et al. describes the identification of a protein “reader” of the histone H3 mark, H3K36me3, the protein PWWP2A. The authors further characterize the complex PWWP2A is found in and determine that it interacts with HDAC1/2, MTA and RBBP4/7 (NuRD complex) and other proteins. The authors further demonstrate that the interactions are dependent on the PWWP domain of the PWWP2A protein. Lastly, the authors show that this protein is found on chromatin that contains H3K36me3, and also to some extent regulates the binding of some of the interacting proteins (i.e. HDAC1) to these same genes. Overall, the experiments are well put together and manuscript for the most part clearly written, and I recommend publication of this manuscript in Nature Communications. Some questions I had were:

1. From the proteomics data in Figure 1, the enrichment of PWWP2A and other interactors such as HDAC2, RBBP7, and HTLF are quite weak, especially compared to MSH6 (the previously known H3K36me3 binder). Were statistical tests (i.e. t-tests or ANOVA) used to show that the enriched proteins were significant? Were these analyses only repeated twice?

Yes, the SILAC nucleosome affinity purifications were performed twice and the labelled proteins were identified as significant outliers by interquartile range analysis.

2. Can the authors better explain why loss of PWWP domain of PWWP2A protein only affects enrichment of gene body versus promoter regions?

PWWP2A contains three characterised domains, encompassing the MTA:HDAC-interacting region (identified in this study), the H2A.Z binding region, and the PWWP domain. This protein is able to bind both H2A.Z which is predominantly enriched at active gene promoters, and H3K36me3 which is found at active gene bodies. Deletion of the PWWP domain affects only PWWP2A’s ability to bind H3K36me3 and gene body regions, but does not seem to affect its H2A.Z-binding and promoter localisation.

3. The ChIP-Seq data in Figure 4 is a little confusing and contradictory to the data presented in Figure 1E. Seems like the genome-wide patterns for H3K36me3, PWWP2A are in the body of the genes, but HDAC1, MTA2, MBD3 are most at the TSS.

Apologies for the confusion, we have added a section in the text to detail why the binding profile of HDAC1 and PWWP2A are so different. Firstly the MTA and HDAC1/2 are highly abundant proteins that are found in different types of HDAC complexes. Quantitative MS of MTA1 interactions in human cells found that 32% of MTA1 interactions with MBD3, 28% with MBD2, and 19% with PWWP2A (Hein Cell 2015). Only a subset of MTA:HDAC associates

with PWWP2A, and by ChIP-seq it appears the majority of MTA and HDAC1/2 proteins are in complexes that are recruited to promoter regions. Secondly, ChIP relies on capture of the protein-chromatin interaction through chemical crosslinking. It is known that chromatin at gene body regions undergo acetylation and deacetylation (Crump PNAS 2011), however the HATs and HDACs are only captured at promoter regions possibly due to increased abundance or more stable binding at promoters compared to gene bodies (Wang Cell 2009). PWWP2A on the other hand which stably interacts with chromatin through H3K36me3, is efficiently captured at gene bodies.

Importantly, we directly test the effect of PWWP2A/B loss on HDAC1 recruitment. Our results in Fig. 5 show that upon deletion of PWWP2A/B, HDAC1 is lost and H3K27ac is increased at the promoter but also over the gene bodies of PWWP2A high occupied genes, suggesting that PWWP2A does recruit a subset of the MTA:HDAC complex to gene bodies as well as promoters.

Reviewer #2 (Remarks to the Author):

In this manuscript, Zhang and colleagues provide a new H3K36me3 interacting factor – PWWP2A/B – identified through a biochemical MS screen. This protein has interesting associations with known chromatin regulators that are probed in biochemical assays. Function in chromatin and gene regulation is pursued in knock-out cell lines using genomic techniques.

I find the first portion of this manuscript – the biochemistry of discovery – to be high quality, compelling and of high interest.

I find the second portion of this manuscript – the functional genomics analysis – to be problematic in many respects which I detail below. I urge the authors to consider the conclusions drawn from the data presented – I find many to be unsupported by the data.

Major questions:

1. I find it disturbing that MTA1/2/3 are not identified in the initial nucleosome screen. If the complex discussed in the text is relevant to K36me3, why is MTA1/2/3 not enriched?

The nucleosome bait captures all chromatin binding proteins, therefore the sample complexity is huge. The mass spectrometer only sequenced the top 15 precursor ions per cycle, and cannot acquire information on all the peptides present in the sample. In our experiment unfortunately no peptides of the MTA1/2/3 were detected by the mass spec. Therefore, it is not that MTA1/2/3 is not enriched, it was just not detected at all.

This leads one to wonder whether the biochemistry provided from 293F cells reflects the species detected by nucleosome binding. (Note that I find the biochemistry from 293F cells to be of high quality). I would really like to see the 'data not shown' of PWWP2 lack of interaction with HDAC1 alone. I would also like to see PWWP2 interaction (or lack thereof) with HDAC1 coexpressed with RBBP7 (recalling that Schreiber's original description of HDAC1 was a 2 protein complex – HDAC with RBBP4/7). The bottom line for me is that I am not certain that the species identified by nucleosome binding and the species identified

biochemically in 293F cells (and by the other co-IP/western experiments) are the same thing. This is central to the story and conclusions. Please help me to understand.

Thank you for the suggestion, we have now added the PWWP2A HDAC co-expression experiment to Supplementary Fig. 3a. Flag-PWWP2A (1-373) was co-expressed with HDAC1, and with RBBP4 and/or MTA1 (162-354). Proteins were purified on Flag resin before cleaving with TEV protease. A) PWWP2A was co-expressed with HDAC1. Only PWWP2A is observed. B) On the addition of MTA1, PWWP2A is able to bind to HDAC1:MTA1 as part of a stoichiometric complex. RBBP4 is not present in this complex. C) Co-expression of PWWP2A, HDAC1 and RBBP4 without MTA1 does not result in a complex i.e. Flag-PWWP2A elutes alone.

With regard to HDAC1 being a 2 protein complex with HDAC and RBBP4/7, please refer to recent crystallographic and EM studies which show that HDAC and RBBP4/7 assemble around the corepressor protein MTA which is critical for complex formation (Millard Mol Cell 2013, Millard eLife 2016).

2. While I certainly applaud the effort to produce DKO cells and do functional genomic experiments, I find those presented here to be hard to understand and certainly think the results do not support the conclusions.

The data around pausing are very problematic for me. The authors show very small changes in the pausing index by metagene plot. More disturbing is the candidate gene shot in 6C. It looks like very marginal changes in pausing index in a highly expressed gene, with big changes in pausing index at a gene expressed at very low levels.

Is the global change indicated in 6a-c driven by large changes in pausing index at barely expressed genes?

We apologise that the candidate genes shown were not representative of the data and have now amended that panel. We performed further analysis which shows that the pausing index is greater at high expressed genes compared to low expressed genes (Supplementary Fig. 6c). Highly expressed genes are also more occupied by PWWP2A which is where we see a greater effect in pausing (Fig. 6c).

To better understand Pol II dynamics and the increase in paused Pol II upon DKOs, we performed Ser5P RNA Pol II calibrated ChIP-seq. Ser5P is enriched on stalled RNA Pol II. In three biological replicates of the two DKO cell lines, we observed significantly increased Ser5P Pol II around the TSS-proximal region of PWWP2A highly occupied genes compared to wildtype (Fig. 6d). Therefore the increase in TSS-proximal total Pol II seen in our first ChIP is due to increased levels of paused Ser5P Pol II at these genes.

The conclusions around changes in K9Ac are likewise problematic for me. I do not believe that the data in S4f support the authors conclusions that K9Ac increases at PWWP target genes.

We have now averaged the biological replicates for all our ChIP-seq data. We agree that increases in H3K9ac are much less pronounced than increases in H3K27ac and have modified the main text to avoid overstating the H3K9ac result.

Perhaps most importantly, I do not see in this data compelling evidence for colocalization in the genome of PWWP2A with HDAC1 or MTA2. The heat map in Figure 4D is not informative – the data for MTA2, HDAC1, MBD3, CHD4 are presented in a completely different manner than PWWP2A and K36me3. Figure 4e seems to me to conclusively demonstrate a lack of colocalization of PWWP2A with HDAC1 (which does colocalize with PWWP2B and K4me3 – but not with K36me3).

I do not understand how the authors reach the conclusion drawn in this section of the manuscript given the data presented.

Apologies for the presentation, we have now replotted the heatmap so the data is presented over the entire genic region for all proteins. We did not mean to imply that the heatmap in Fig. 4 demonstrates that PWWP2A and HDAC1 colocalise, and we have changed the text to be clearer.

Please refer to our response to Reviewer 1 major question 3 where we elaborate in detail on these points.

Minor questions:

1. The recombinant nucleosome fishing experiment is very interesting and I like the approach. The validation by id of a known K36me3 binding factor is nice. I wonder about proteins excluded from K36me3 – LUC7L1, TBL2, ... - can the authors provide any rationale why these proteins should be excluded from modified nucleosomes?

We did not pursue or further characterise any of the proteins that were 'repelled' by H3K36me3 for the purposes of the current study. To speculate, chromatin binding domains which recognise specific histone tails residues are often affected by modifications, either positively or negatively. We surmise that these proteins may bind nucleosomes near the H3K36 residue and binding affinity is decreased in the presence of H3K36 methylation.

2. I find the gel filtration chromatography experiment less than compelling for clear demonstration of separate complexes. I suggest the authors explore a different matrix – perhaps ion exchange – that will give baseline resolution of PWWP and its pairing partners with conventional NuRD complex.

Unfortunately, none of these protein complexes survive ion exchange chromatography. We have tried to the best of our ability to demonstrate using nuclear extracts (Fig. 2) and by in vitro reconstitution (Fig. 3) that they form separate complexes. And as we have mentioned in the discussion section, PWWP2A/B and MBD2/3 have never been shown to co-purify with each other in previous MS studies.

3. The co-IP experiments with FLAG (2d) would benefit from negative controls (i.e. PWWP 1-148, PWWP 373-649)

Thank you for this suggestion, we have now added a new truncation 383-755 which does not interact with MTA1 or HDAC1.

4. Methods for chip in ES cells could be more detailed. Reference to a previous paper requires the reader to search for how the experiment was performed. Please provide the methods here.

We have now included a detailed outline of how calibrated native and crosslinking ChIP were performed.

5. Please validate expression of MTA2 and HDAC1 in the DKO cells. Does loss of all PWWP2A/B result in decreased expression of these proteins?

The levels of NuRD protein subunits were not visibly affected by western in the DKO cells compared to wildtype (Supplementary Fig. 5e).

6. Figure 5e is very difficult to follow. Please consider a different presentation.

We recognise that this figure is very busy, but we have tried several different presentation styles and colour combinations. In the end this presentation with overlaid tracks is the best visualisation of the data. Furthermore we did want to show that there is consistency between our replicates.

Reviewer #3 (Remarks to the Author):

The authors use a quantitative proteomic approach to identify protein complexes that bind H3K36me3, leading to the discovery of a new variant of the NuRD complex. They verify the exclusive choice between components specific to the two variants (previously known one and the newly identified one) using orthogonal biochemical assays. They then use ChIP to identify the genomic location of the NuRD complex variants and use genetic perturbations to dissect their potential function.

Discussion:

This manuscript can be roughly divided to two parts; the first is a biochemical study of protein complexes associated with H3K36me3 and with their binders. The second is a genomic study of the impact of deletion of two of the binding proteins PWWP2A/B on chromatin marks and transcription. The first part provides clear and compelling evidence for the role of PWWP2A/B in NuRD complex formation and their potential function in chromatin binding. The second part is more complex and less clear. The analysis of localization of NuRD complex components and the effect of double KO of PWWP2A/B on chromatin marks identifies clear pattern which are consistent with the biochemical results (binding preferences of PWWP2A/B and HDAC activity of NuRD). From these, however, it is hard to infer a mechanistic role of these NuRD variants in transcription. The PolIII/4sU analysis is sketchy and less compelling (see comments below). Thus, it is hard to understand whether the changes in PolIII pausing are due to missing deacetylation step in the DKO or due to various

potential indirect effects (some of which are discussed in lines 354-362).

My recommendation is either to remove the last part and publish the solid aspect of the results, or to strengthen the results in this last part with proper analysis to understand whether the effects are due to changes in acetylation levels or to missing activity of the PWWP2-MTA-HDAC complex. Comparison to global HDAC inhibition will further define the specific contribution of the PWWP2 complexes in these effects.

Comments:

* Double KO (Fig 5) – why such discrepancies in the signal of the double KO ChIP?

We used different CRISPR guides in the generation of the PWWP2A deletion (please see Supplementary Fig. 5) to generate two completely independently derived DKO lines. Although the NuRD-interacting and chromatin binding regions of PWWP2A/B are removed in both DKO clones, exact deletions themselves are very different. Aside from possible different CRISPR associated off target effects, or the nature of the deletions themselves, these cells have undergone selection pressure and passaging so may have acquired other clonal differences. We have performed multiple ChIP-seq replicates for each DKO, and while the absolute values may differ they do show a consistent trend for all our experiments from characterisation of HDAC recruitment, histone acetylation, and pausing analysis.

* Changes in K9/27ac – clearly the regions of acetylation are of varying length. I am not sure if total ChIP signal is the right measure to look at the distribution. The dynamic range in the distribution among genes (targets/non-targets) is much larger than the differences between strains. If there are confounding factors (gene length, expression levels, etc) a paired analysis (ratio of change per gene between DKO and WT) will provide a tighter and more informative distribution.

We appreciate the suggestions. PWWP2A binds to all active genes (see heatmap Fig. 4d), and we defined PWWP2A targets or high occupancy genes as the top 10% of PWWP2A bound genes. However the majority of remaining genes are still bound by PWWP2A to a lesser extent. Both groups show changes in DKO cells, but high occupancy genes show a more dramatic effect. To clarify this, we have changed the categories from targets/non-targets to high and low occupied genes.

Because changes in acetylation occurred across the entire gene (especially for H3K27ac), we looked at the total signal, where we observe greater increases in acetylation at genes which are highly occupied by PWWP2A (Fig. 5e and Supplementary Fig. 5e). To compare regions of the same length, we plotted the meta-gene profile for H3K9ac and H3K27ac \pm 5kb of the TSS for PWWP2A high and low occupancy genes and observe the same effect as for the total signal (Supplementary Fig. 5g).

* Analysis of PolII pausing is done in a brief and unconvincing manner. In the analysis of Fig4 the authors used spike-ins to normalize the signal (“calibrate ChIP” in their methods). They do not use similar calibration in the analysis of Pol II signal. The immediate question that comes to mind is whether there are significant changes in total amount of active/paused

transcription in the DKO strains. Since NuRD, like most transcriptionally related machinery, is generic and has effects on all (or most) actively transcribed genes, one has to be careful in analysis of signal without proper study of absolute levels. For example, the global increase of acetylation levels might lead to sequestering of PolII at many abnormal sites, leading to reduction of PolII at bona-fide transcripts. Alternatively, cell might compensate and generate higher levels of PolII leading to accumulation of paused PolII at gene starts (traffic jams) while the actual levels of productive transcriptions remain more or less constant. Without calibration changes in the relative signal are hard to interpret, and various potential models (as the above examples) might provide “reasonable” account of the phenomena.

Thank you for all of your suggestions. First the global levels of total Pol II, Ser5P and Ser2P Pol II do not appear to greatly differ by western blot between wildtype and DKO cells (Supplementary Fig. 6e). To better understand the dynamics of the increased promoter-proximal Pol II, we performed three biological replicates of calibrated ChIP-seq for the Ser5P and Ser2P phosphorylated forms of Pol II. Ser5P Pol II is statistically significantly increased at the TSS-proximal region in the DKO cells, while Ser2P is not significantly different. Therefore the increase in total Pol II that we observe in the TSS of DKO cells is due to increased levels of paused Ser5P Pol II. It may be that increasing levels of histone acetylation upon loss of HDAC recruitment following PWWP2A/B deletion causes increased Pol II initiation at active genes causing a build-up of stalled Ser5P Pol II at these sites.

* 4sU-seq analysis – I really can’t tell what is shown in Supp Fig 4i and how it “shows a similar trend”

We amended the figure, and inserted the meta-gene profile for 4sU-seq on the PWWP2A highly occupied genes across the TSS \pm 500bp region, shown in Suppl. Fig. 6b. By “shows a similar trend”, we mean a slight increase in TSS-proximal nascent transcription may be a result of the increase in TSS-proximal Pol II that we see.

Minor issues:

Fig 1c – I am curious why there is a substantial correlation between H/L in the two experiments? The non-specific signals seem as strong as the actual signal.

The non-specific signals are due differences between the extraction of proteins in the H and L samples. The heavy and light labelled cells were grown and harvested on different days, therefore there will be slight differences in extraction due to technical variation. After consulting another group that routinely does SILAC labelling, we were told that technical variation can be reduced by preparing samples in small batches and mixing them together for the experiment.

Fig 4c – I did not find this figure especially clear nor informative on top of the very clear signal we see in Figs 4a and b.

We wanted another graphic way to show that PWWP2A/B, as well as other NuRD complex components are targeted to active genes.

Line 238 “data now shown”?

Thanks for the correction. We have now included the images of embryoid body formation in Supplementary Fig. 5c.

Fig 5 – key of the color code on the figure itself will help the reader.

Thanks for the suggestion. The key of the color code was added.

Fig 5e – the overlay is hard to read. In the top tracks the transparent red/blue are on top of the gray while on the bottom tracks they are obscured by the gray track. I appreciate that this is a lot of information to convey. The technical repeats are very consistent, which is great, but showing all of them here (rather than a sup fig) creates information overload that takes from the main message of the figure.

We recognise that this figure is very busy, but we have tried several different presentation styles and colour combinations. In the end this presentation with overlaid tracks is the best visualisation of the data. Furthermore we did want to show that there is consistency between our replicates.

Fig 5f – use consistent labels (e.g., E14 vs WT etc), preferably an informative one (DKO1 rather than C7)

We have changed all our labels to be consistent. Wildtype is now WT, but we have decided to use the C7 and A1 to represent the two knockout lines which allows us to process experiments and raw data with greater ease.

Reviewers' comments:

Reviewer #1 (Remarks to the Author):

I am satisfied with the responses to the questions I had in the initial review (Reviewer #1). However, I feel less confident that the authors have fully responded to all of the questions of the rest of the reviewers, especially Reviewer #2.

Reviewer #2 (Remarks to the Author):

I thank the team for compiling a largely responsive answers to previous questions about this interesting story.

I still find the gel filtration to be less than compelling in convincing me of the existence of two biochemical entities. I once again encourage the use of alternative chromatography matrices OR just drop this data as supporting the argument presented.

Editorial Note: Reviewer #3 was unable to provide comments for the revised manuscript and a fourth Reviewer was therefore recruited to look over the authors' rebuttal.

Since the initial submission of this manuscript, we discovered an error in the pipeline of our calibrated ChIP-sequencing analysis. A scale-down parameter used for conventional analysis was not properly converted for use in calibrated ChIP-analysis resulting in the application of improper normalisation. Our scripts and the calibration parameters used in this study can be found at github.com and Supplementary Table 5.

This does not affect the experiments which were processed through conventional ChIP-seq analysis (all the experiments in Fig. 4, and the total Pol II ChIP in Fig. 5).

Calibrated ChIP was performed in 3 experiments to compare the profiles of histone acetylation, MTA:HDAC binding, and the phosphorylated forms of Pol II in WT and DKO cells. After reanalysing these three calibrated ChIP-seq experiments, the data shows the following.

- The levels of H3K9ac and H3K27ac are significantly increased in DKO cells compared with WT, most notably around the TSS proximal region but also over the rest of the gene.
- MTA2 and HDAC1 binding are not significantly different between WT and DKO cells.
- The relative increase in TSS-proximal pausing we see in the total Pol II ChIP (conventional ChIP) upon PWWP2A/B DKO is due to a decrease in the elongating Ser2P Pol II. The binding profile of Ser5P Pol II is not significantly different between WT and DKO.
- After suggestions from the reviewers, we have performed further analysis to look at the groups of genes most affected by PWWP2A/B DKO, and we consistently observe that genes with high PWWP2A occupancy, and high levels of gene expression show the greatest changes upon loss of PWWP2A/B.

We thank the reviewers for their thoughtful and helpful comments. Our responses are detailed below.

Reviewer #2 (Remarks to the Author):

I thank the team for compiling a largely responsive answers to previous questions about this interesting story.

I still find the gel filtration to be less than compelling in convincing me of the existence of two biochemical entities. I once again encourage the use of alternative chromatography matrices OR just drop this data as supporting the argument presented.

Thank you for comments. We want to emphasise that the gel filtration experiment in Fig. 3 is not the only evidence supporting the existence of two mutually exclusive complexes. Mass spectrometry data from our work and the literature strongly supports mutual exclusivity of

PWWP2A and MBD3 complexes. We have performed three IP-MS experiments to determine PWWP2A/B interactors in human and mouse cells and have never identified a single peptide of MBD2/3, CHD3/4, or p66 α/β , despite being able to greatly enrich for MTA1/2/3, HDAC1/2, and RBBP4/7. Furthermore, IP-MS of MBD3 interactors have never identified PWWP2A/B as interactors (Smits et al. 2013, Kloet et al. 2015, supplementary MS data), and only find canonical NuRD complex components. Considering that both PWWP2A and MBD3 co-purify the same interaction partners (i.e. MTA, HDAC, RBBP), the fact that these reciprocal IP-MS studies show that PWWP2A and MBD3 never co-purify one another, suggest that they do not co-exist in the same complex even in small amounts. We believe that these mass spectrometry experiments taken together with reconstitution of the two complexes in vitro offer compelling evidence suggesting the existence of two mutually exclusive complexes that assemble around an MTA:HDAC subcomplex. Thus, we believe that the current data is sufficient for us to suggest our model for the assembly of NuRD and variant NuRD complexes. Ultimately, structural studies will be required to unequivocally prove this point. We have reworded the text so that it is clear that all the evidence taken together and not just the gel filtration experiment presented in Fig. 3 lead us to suggest the mutual exclusivity model.

Smits, A.H., Jansen, P.W., Poser, I., Hyman, A.A. & Vermeulen, M. Stoichiometry of chromatin-associated protein complexes revealed by label-free quantitative mass spectrometry-based proteomics. *Nucleic Acids Res* **41**, e28 (2013).

Kloet, S. L. *et al.* Towards elucidating the stability, dynamics and architecture of the nucleosome remodeling and deacetylase complex by using quantitative interaction proteomics. *FEBS J* **282**, 1774-1785 (2015).

Reviewer #3 (Remarks to the Author):

The authors use a quantitative proteomic approach to identify protein complexes that bind H3K36me3, leading to the discovery of a new variant of the NuRD complex. They verify the exclusive choice between components specific to the two variants (previously known one and the newly identified one) using orthogonal biochemical assays. They then use ChIP to identify the genomic location of the NuRD complex variants and use genetic perturbations to dissect their potential function.

Discussion:

This manuscript can be roughly divided to two parts; the first is a biochemical study of protein complexes associated with H3K36me3 and with their binders. The second is a genomic study of the impact of deletion of two of the binding proteins PWWP2A/B on chromatin marks and transcription. The first part provides clear and compelling evidence for the role of PWWP2A/B in NuRD complex formation and their potential function in chromatin binding. The second part is more complex and less clear. The analysis of localization of NuRD complex components and the effect of double KO of PWWP2A/B on chromatin marks identifies clear pattern which are consistent with the biochemical results (binding preferences of PWWP2A/B and HDAC activity of NuRD). From these, however, it is hard to infer a mechanistic role of these NuRD variants in transcription. The PolIII/4sU analysis is sketchy and less compelling (see comments below). Thus, it is hard to understand whether the changes in PolIII pausing are due to missing deacetylation step in the DKO or due to various potential indirect effects (some of which are discussed in lines 354-362).

My recommendation is either to remove the last part and publish the solid aspect of the results, or to strengthen the results in this last part with proper analysis to understand whether the effects are due to changes in acetylation levels or to missing activity of the PWWP2-MTA-HDAC complex. Comparison to global HDAC inhibition will further define the specific contribution of the PWWP2 complexes in these effects.

Thank you for your suggestion, we agree that the biochemical characterisation is more solid and that the second half of the manuscript is more complex. However, as this marks the first time that the PWWP2A/B variant NuRD complex has been biochemically characterised, we wanted to gain some insight into its functions in a cellular context as well. We have further characterised the effect of PWWP2A/B deletion on RNA Pol II binding and the relative increase in Pol II pausing through more detailed bioinformatics analysis as well additional experiments.

Comments:

* Double KO (Fig 5) – why such discrepancies in the signal of the double KO ChIP?

We used different CRISPR guides in the generation of the PWWP2A deletion (please see Supplementary Fig. 5a,b) to generate two completely independently derived DKO lines. Although the NuRD-interacting and chromatin binding regions of PWWP2A/B are removed in both DKO clones, exact deletions themselves are very different. Aside from possible different CRISPR associated off target effects, or the nature of the deletions themselves, these cells have undergone selection pressure and passaging so may have acquired other clonal differences. We have performed multiple ChIP-seq replicates for each DKO, and while the absolute values may differ they do show a consistent trend for all our experiments from analysis of histone acetylation and Pol II.

* Changes in K9/27ac – clearly the regions of acetylation are of varying length. I am not sure if total ChIP signal is the right measure to look at the distribution. The dynamic range in the distribution among genes (targets/non-targets) is much larger than the differences between strains. If there are confounding factors (gene length, expression levels, etc) a paired analysis (ratio of change per gene between DKO and WT) will provide a tighter and more informative distribution.

We appreciate the suggestions. PWWP2A binds to all active genes (see heatmap Fig. 4d), and we initially defined PWWP2A targets or high occupancy genes as the top 10% of PWWP2A bound genes. However the majority of remaining genes are still bound by PWWP2A to a lesser extent. Both groups show changes in DKO cells, but high occupancy genes show a more dramatic effect. To clarify this, we have changed the categories from targets/non-targets to high and low occupancy genes.

We observe greater increases in acetylation at genes which are highly occupied by PWWP2A (Fig. 5a-d). Increases in acetylation also correlates with the gene expression level as high expressed genes gain more acetylation than intermediate and low expressed genes

upon PWWP2A/B deletion (Supplementary Fig. f,g). This is expected as PWWP2A/B occupancy is positively correlated with gene expression.

To compare regions of the same length, we compared H3K9ac and H3K27ac signal over the region $\pm 5\text{kb}$ of the TSS for PWWP2A high and low occupancy genes (Fig. 5a-d), and find that both increase over this region. As well, we compare the acetylation in gene body region for all genes after normalising for gene length (Supplementary Fig. 5h,i), and find that there is a small but significant increase in H3K9ac over gene bodies as well.

* Analysis of PolII pausing is done in a brief and unconvincing manner. In the analysis of Fig4 the authors used spike-ins to normalize the signal (“calibrate ChIP” in their methods). They do not use similar calibration in the analysis of Pol II signal. The immediate question that comes to mind is whether there are significant changes in total amount of active/paused transcription in the DKO strains. Since NuRD, like most transcriptionally related machinery, is generic and has effects on all (or most) actively transcribed genes, one has to be careful in analysis of signal without proper study of absolute levels. For example, the global increase of acetylation levels might lead to sequestering of PolII at many abnormal sites, leading to reduction of PolII at bona-fide transcripts. Alternatively, cell might compensate and generate higher levels of PolII leading to accumulation of paused PolII at gene starts (traffic jams) while the actual levels of productive transcriptions remain more or less constant. Without calibration changes in the relative signal are hard to interpret, and various potential models (as the above examples) might provide “reasonable” account of the phenomena.

Thank you for all of your suggestions. First the global levels of total Pol II, Ser5P and Ser2P Pol II do not appear to greatly differ by western blot between wildtype and DKO cells (Supplementary Fig. 6g). To better understand the changes in Pol II pausing, we performed calibrated ChIP-seq for the Ser5P and Ser2P phosphorylated forms of Pol II. Ser5P Pol II does not appear to be significantly changed in DKO cells, while Ser2P Pol II is decreased along the gene body as well as downstream of transcriptional terminal site (Fig. 5h,i). This is more pronounced at PWWP2A high occupancy genes.

We infer that the relative increase in RNA Pol II pausing observed by total Pol II ChIP-seq upon PWWP2A/B deletion is due to decreased Ser2P Pol II indicating an elongation defect.

* 4sU-seq analysis – I really can’t tell what is shown in Supp Fig 4i and how it “shows a similar trend”

We amended the figure, and inserted the meta-gene profile for 4sU-seq on the PWWP2A highly occupied genes across the TSS $\pm 500\text{bp}$ region, shown in Supplementary Fig. 6d,e. We observe that upon loss of PWWP2A/B, there is an increase in the TSS-proximal read density relative to gene body read density, which is similar to there being a more promoter-proximal compared with gene body Pol II.

Minor issues:

Fig 1c – I am curious why there is a substantial correlation between H/L in the two experiments? The non-specific signals seem as strong as the actual signal.

The non-specific signals are due to differences between the extraction of proteins in the H and L samples. The heavy and light labelled cells were grown and harvested on different days, therefore there will be slight differences in extraction due to technical variation. After consulting another group that routinely does SILAC labelling, we were told that technical variation can be reduced by preparing samples in small batches and mixing them together for the experiment.

Fig 4c – I did not find this figure especially clear nor informative on top of the very clear signal we see in Figs 4a and b.

We wanted another graphic way to show that PWWP2A/B, as well as other NuRD complex components are targeted to active genes.

Line 238 “data now shown”?

Thanks for the correction. We have now included the images of embryoid body formation in Supplementary Fig. 5c.

Fig 5 – key of the color code on the figure itself will help the reader.

Thanks for the suggestion. The key of the color code was added.

Fig 5e – the overlay is hard to read. In the top tracks the transparent red/blue are on top of the gray while on the bottom tracks they are obscured by the gray track. I appreciate that this is a lot of information to convey. The technical repeats are very consistent, which is great, but showing all of them here (rather than a sup fig) creates information overload that takes from the main message of the figure.

We recognise that this figure is very busy and difficult to follow, and therefore decided not to show it. Instead, in the meta-gene plot we averaged the signal from biological replicates, which are calibrated and reproducible.

Fig 5f – use consistent labels (e.g., E14 vs WT etc), preferably an informative one (DKO1 rather than C7)

We have changed all our labels to be consistent. Wildtype is now WT, but we have decided to use the C7 and A1 to represent the two knockout lines which allows us to process experiments and raw data with greater ease.

REVIEWERS' COMMENTS:

Reviewer #2 (Remarks to the Author):

In this revision, the authors provide new, corrected analysis of chip seq data.

I find the lack of colocalization of MTA2 and HDAC1 with PWWP over gene bodies (Figure 4d) to be contrary to the conclusions drawn by the authors.

I find the lack of a significant change in localization of MTA2 and HDAC1 over gene bodies in the double PWWP KO cells to be contrary to the model derived by the authors.

The biochemistry and chip data are not in agreement here. While I am reluctant to ask the authors to produce a specific result, I question whether the model is correct - that PWWP directs localization of NuRd components to gene bodies to regulate acetylation of histones.

Reviewer #4 (Remarks to the Author):

The Authors have convincingly replied to the referees comments. Both parts of the manuscript contain enough of individual pieces of data to support the model presented in Figure 6.

However, I am not convinced that treating the two proteins - PWWP2A and PWWPAB - as one complex with entirely overlapping functions is appropriate given that they show different localization patterns (one along gene body and the other on promoters and enhancers). There should therefore be some extra discussion aimed at putting all the observations about the two studied proteins into one coherent picture.

Given that there is quite some discrepancy between the two DKO lines the Authors should limit the data to those where the two lines indeed show the same trend. I am not convinced about the effect of the DKO on the H3K9ac and the H3K27 along the gene body. Perhaps it is better to remove this piece of data from the manuscript.

Response to Reviewers' comments:

Reviewer #2 (Remarks to the Author):

In this revision, the authors provide new, corrected analysis of chip seq data.

I find the lack of colocalization of MTA2 and HDAC1 with PWWP over gene bodies (Figure 4d) to be contrary to the conclusions drawn by the authors.

I find the lack of a significant change in localization of MTA2 and HDAC1 over gene bodies in the double PWWP KO cells to be contrary to the model derived by the authors.

The biochemistry and chip data are not in agreement here. While I am reluctant to ask the authors to produce a specific result, I question whether the model is correct - that PWWP directs localization of NuRd components to gene bodies to regulate acetylation of histones.

Thank you for your comments and for taking the time to look over our manuscript again. We addressed the lack of colocalisation of between MTA2 and HDAC1 with PWWP2A in detail to reviewer 1 in the first response. Briefly, the majority of MTA2 and HDAC1 are found in complexes not containing PWWP2A, and are targeted to the promoter regions. Only a fraction of MTA2/HDAC1 coexist with PWWP2A, therefore the binding profile of MTA2/HDAC1 by CHIP largely shows promoter localisation. We highlighted the section in the main text where we discuss the differences between the CHIP profiles.

Our biochemical studies indicate that PWWP2A binds H3K36me3 and forms a stable and catalytically active HDAC complex, suggesting that it could be a means to regulate gene body deacetylation. However we show that PWWP2A also has promoter binding albeit to a lower degree than gene body binding, suggesting that it may regulate histone acetylation over promoters and gene bodies. We agree with you that by CHIP-seq there is a lack of changes in MTA2 and HDAC1 binding upon PWWP2A/B DKO, which we believe may be due to CHIP not being sensitive enough to capture small changes in binding or changes in binding dynamics upon PWWP2A/B loss. We have amended the text to avoid overspeculating about the role of PWWP2A in gene body deacetylation.

Reviewer #4 (Remarks to the Author):

The Authors have convincingly replied to the referees comments. Both parts of the manuscript contain enough of individual pieces of data to support the model presented in Figure 6. However, I am not convinced that treating the two proteins - PWWP2A and PWWPAB - as one complex with entirely overlapping functions is appropriate given that they show different localization patterns (one along gene body and the other on promoters and enhancers). There should therefore be some extra discussion aimed at putting all the observations about the two studied proteins into one coherent picture.

Thank you for your comments. We agree that PWWP2A and PWWP2B have many overlapping functions yet cannot be considered entirely redundant, especially with differences in their localisation patterns to chromatin as you pointed out. As the two paralogs can coexist in the same HDAC complex, we decided to generate a DKO first, however future studies with single KOs are needed to address paralog-specific functions. We have now added this as a discussion point in our manuscript.

Given that there is quite some discrepancy between the two DKO lines the Authors should limit the data to those where the two lines indeed show the same trend. I am not convinced about the effect of the DKO on the H3K9ac and the H3K27 along the gene body. Perhaps it is better to remove this piece of data from the manuscript.

Both DKO lines show increases in H3K9ac along the gene body but only one clone shows increases in H3K27ac along the gene body. We would prefer to keep the data in the Supplementary Figures but we have changed the text in the manuscript to avoid overstating this result.